# Graph Transformer based Large Neighborhood Search via Expert Guided Reinforcement Learning

## Abstract

Mixed-Integer Linear Programs (MILPs) have wide-ranging applications across various fields. Recently, significant research efforts have been directed towards developing learning-based Large Neighborhood Search (LNS) methods for efficiently identifying high-quality MILP solutions. Most existing works focus on imitation learning, which faces two major challenges: (i) the performance is limited by the expert policy itself, and (ii) learning the graph representation of MILPs cannot effectively explore the global graph structure due to the limited receptive field of Graph Convolutional Networks (GCNs). To address these issues, we propose a novel expert-guided reinforcement learning model to design the destroy operator in LNS. In our approach, the expert provides weighted guidance to assist the learning agent efficiently. Additionally, we introduce a novel graph transformer-based network, which captures both local and global information efficiently. We prove that our graph transformer-based network is more expressive than 1-WL test and can distinguish non-isomorphic MILP graphs successfully. We conduct extensive evaluations to demonstrate the effectiveness of our proposed algorithm, showing significant improvements in the performance of LNS for MILPs. The code and models will be made publicly available at https://anonymous.4open.science/r/Graph-Transformer-based-LNS-822F/README.md.

## 1 Introduction

Mixed-integer Linear Programming (MILP) problems, a special subset of Combinatorial Optimization (CO) problems, have been widely applied in various fields such as transportation, production planning, and resource allocation (Toth & Vigo, 2002; Paschos, 2014). A MILP problem usually includes both continuous and discrete variables, and is generally NP-hard. Due to its discrete and non-convex feasible region, it's challenging to design optimization methods. Researchers have proposed various techniques and algorithms to solve MILP problems. One classic method is Branch-and-Bound (B&B) (Land & Doig, 1960), which accurately solves MILP problems by decomposing them into a series of sub-problems and reducing the search space by limiting the range of variable values. It requires extensive trial-and-error and significant domain knowledge, leading to high costs and low efficiency. Therefore, much research effort has been dedicated to improve the efficiency of solving MILP problems.

Applying machine learning methods to solve MILP problems has become a promising research direction. Alvarez et al. (2014) were the first to apply machine learning to branching variable selection, using classical supervised learning models to approximate strong branching decisions. To reduce complex feature calculations, the GCNN algorithm (Gasse et al., 2019) models the MILP problem as a bipartite graph and uses a graph convolutional network (GCN) (Kipf & Welling, 2017) to obtain features. This approach has become the dominant representation method for MILP and has been followed by many subsequent works. However, these methods rely on B&B and thus have high complexity, which limits their application in difficult MILP problems. Recently, the Large Neighborhood Search (LNS) method, which destroys and repairs the current solution to achieve continuous improvements, has proven more suitable for large-scale mixed-integer optimization. For instance, ND+NNS (Sonnerat et al., 2022) learns the destroy operator in LNS by imitating Local Branching (LB) (Fischetti & Lodi, 2003). Building on this work, CL-LNS (Huang et al., 2023) collects

positive and negative solution samples from LB and learns an efficient operator using contrastive learning. In this paper, we also adopt the LNS framework to solve the difficult MILP problems.

There are two main limitations in the dominant methods adopting the LNS framework. First, previous works often rely on GCN-based methods to learn representations from bipartite graphs due to their strong performance in graph learning. However, the graph convolution in GCNs functions as a localized filter. Although additional layers of GCNs can capture a broader view of the graph, they suffer from an over-smoothing issue, where all node representations converge to a constant after enough layers (Li et al., 2018; Zhang et al., 2024). In MILP problems, GCN-based methods with limited layers struggle to capture these complex global relationships among variables and constraints. In addition, they cannot distinguish non-isomorphic MILP graphs due to their upperbounded expressive ability over the 1-WL test as stated by Chen et al. (2023). Second, most research focuses on imitation learning, which requires experts to generate solution samples, thus limiting the learning performance. Reinforcement learning (RL) methods, which learn the destroy policy for the destroy operator from repaired solutions by solving the sub-MILP problem, have the potential to overcome expert limitations. Only one recent work, proposed by Wu et al. (2021), adopts DDPG to learn the destroy policy in LNS. However, RL algorithms often face challenges with unstable convergence and low sampling efficiency during training (Paduraru et al., 2021). Samples from different MILPs can vary significantly in distribution depending on the size and complexity of the MILPs, exacerbating the instability. Additionally, generating each sample requires solving the sub-MILP problem through a solver within the environment, which is time-consuming and leads to an unacceptably high time cost for convergence.

In this paper, we propose a novel expert-guided reinforcement learning model to optimize the destroy operator within the LNS framework. We formulate the LNS process as a Markov Decision Process (MDP) and train the agent to learn an optimal policy for the destroy operator. To address representation challenges, we introduce a novel graph transformer framework featuring two global attention units with linear complexity and a graph convolution layer employing two interleaved half-convolutions. We design a variable-pair (or constraint-pair) attention unit to enable global interactions between arbitrary variable (or constraint) node pairs, while the interaction between variables and constraints is captured through the interleaved half-convolutions. To tackle issues related to instability and low sampling efficiency, we first develop a stable actor-critic framework that separates the actor's policies for each variable independently while evaluating all variables collectively using a critic network. We then introduce an additional expert that provides online weighted guidance during training based on the PPO algorithm. The expert's policy is efficiently learned through a novel weighted imitation learning method, which fully leverages redundant solutions from the solver expert. This expert-guided policy provides weighted guidance for different samples, with stronger guidance applied to samples that perform poorly under the actor's policy and less guidance for those with good performance.

In summary, this paper has the following contributions:

- We propose a novel actor-critic framework to optimize the destroy operator in LNS, with a weighted expert-guided training method based on the PPO algorithm.

- We introduce a novel graph transformer-based network which captures both local and global information efficiently. We prove that it is more expressive than the 1-WL test and can distinguish non-isomorphic MILP graphs successfully.

- We conduct evaluations on three datasets, and the experimental results demonstrate the effectiveness of our proposed algorithm.

## 2 Related Work

Algorithms for solving MILP problems can be roughly divided into two categories: exact solution algorithms and approximate solution algorithms. Exact solution algorithms such as Branch-and-Cut (Land & Doig, 1960) provide the optimal solution for MILP problems. In contrast, approximate solution algorithms, such as LNS, approximate the optimal solution via an iterative way.

## 2.1 Exact solution algorithms

Branch-and-Cut (Land & Doig, 1960) and Decomposition are two widely used algorithms for exactly solving MILP problems.

Branch-and-Cut adds cutting planes to tighten the feasible region of the optimal solution on the basis of B&B, a widely used tree search method to obtain a smaller search tree. Two important parts of optimizing B&B are branching variable selection and node selection. Typical branching variable selection rules include Strong Branching (SB) (Applegate et al., 1995), Pseudo Cost Branching (PB) (Benichou et al., 1971), and Hybrid Branching (HB) (Achterberg & Berthold, 2009). Node selection rules include Breath First Search (BFS) and Depth First Search (DFS). As mentioned before, B&B can be combined with the cutting plane, which constitutes one of the most commonly used methods in modern solvers. Many families of valid inequalities developed, such as Chvatal-Gomory cuts (IPs) and Gomory's fractional cuts. (Zhang et al., 2022) Recently, learning based methods have been applied to optimize the Branch-and-Cut algorithm, focusing on branch variable selection (Sun et al., 2020; Gupta et al., 2020; Shen et al., 2021) and node selection (He et al., 2014; Yilmaz & Yorke-Smith, 2021).

When facing large scale MILP problem, it is necessary to decompose it and then using Branch-and-Cut algorithm to solve it. The main decomposition methods include lagrangian relaxation (Geoffrion, 1974), Benders decomposition (Benders, 2005) and Dantzig-Wolfe decomposition (Dantzig & Wolfe, 1960). Due to the great potential of decomposition based solvers in solving specific structural problems and parallel computing, many works have considered using machine learning methods to rank decomposition (Xavier et al., 2019; Li & Wu, 2022; Hutter et al., 2011).

## 2.2 Large Neighborhood Search

Although Branch-and-Cut is a widely used method for solving MILP problems, it is challenging in many cases, especially for solving large-scale problems. Approximate solution methods that obtain high-quality solutions fast are more suitable for industrial applications. Approximate solution methods can obtain feasible solutions in the early stages of computation and iteratively improve existing solutions.

LNS is an approximate solution method with great potential for solving large-scale problems. Based on some neighborhood definitions, such as fixed variables and added constraints, it defines a relatively large neighborhood of the current solution and generates a sub-problem. Then, various methods are applied to search within the neighborhood to find a better solution. LNS includes LB, Relaxation Induced Neighborhood Search (RINS) (Danna et al., 2005), Relaxation Forced Neighborhood Search (RENS) (Berthold, 2007), Dins (Ghosh, 2007), and so on. When combining machine learning methods, some works try to learn neighborhood selection strategies (Song et al., 2020; Wu et al., 2021), while others focus on automatically finding suitable neighborhood sizes (Liu et al., 2022).

# 3 Background

## 3.1 Mixed-integer Linear Programming

Mixed-integer linear programming problem is a specially case of combinatorial optimization problem. It consists of both continuous and discrete variables with non-convex constraints and is typically defined as:

$$\min \mathbf{c}^\top \mathbf{x}, \mathbf{A}\mathbf{x} \leq \mathbf{b}, \mathbf{l} \leq \mathbf{x} \leq \mathbf{u}, \mathbf{x} \in \mathbb{Z}^n \times \mathbb{R}^p, \tag{1}$$

where $\mathbf{c} \in \mathbb{R}^{n+p}$ is the objective coefficient vector, $\mathbf{A} \in \mathbb{R}^{m \times (n+p)}$ the constraint coefficient matrix, $\mathbf{b} \in \mathbb{R}^m$ the constraint right-hand-side vector, $\mathbf{l}, \mathbf{u} \in \mathbb{R}^{n+p}$ the lower and upper variable bound vectors, $n$ the number of integer variables and $p$ the number of continuous variables. A complete solver tries to solve MILP with the optimal solutions that minimize the objective $\mathbf{c}^\top \mathbf{x}$ under non-convex constraints.

We model the MILP problem as a bipartite graph $\mathcal{G} = (\mathcal{V}, \mathcal{C}, \mathcal{E})$. $\mathcal{V} = \{v_1, v_2, \ldots, v_n\}$ denotes the set of variable nodes on one side of the graph with the feature matrix $\mathbf{Z} \in \mathbb{R}^{n \times d_v}$ and each node $v_i$ represents a variable $x_i$ of the MILP problem. $\mathcal{C} = \{c_1, c_2, \ldots, c_m\}$ denotes the set of constraint nodes on the other side

of the graph with the feature matrix $\mathbf{C} \in \mathbb{R}^{m \times d_c}$ and each node $c_i$ represents a constraint $\sum_j a_{i,j} x_j \leq b_i$ of the MILP problem. An edge $(v_i, c_j) \in \mathcal{E}$ means $a_{i,j} \neq 0$ and $\mathbf{E} \in \mathbb{R}^{m \times n \times d_e}$ denotes the sparse tensor of edge features.

### 3.2 Large Neighborhood Search

LNS is a type of improvement heuristics, which iteratively optimizes a solution by the destroy and repair operators. Specifically, the destroy operator breaks part of the solution $\mathbf{x}_t$ at step $t$. We denote the index of the destroy variables as $\mathcal{N}(\mathbf{x}_t)$. Then, the repair operator fixes the broken solution to derive the next solution $\mathbf{x}_{t+1}$ via solving the following sub-MILP problem:

$$\mathbf{x}_{t+1} = \arg\min_{\mathbf{x}} \{\mathbf{c}^\top \mathbf{x} \mid \mathbf{A}\mathbf{x} \leq \mathbf{b}; \mathbf{l} \leq \mathbf{x} \leq \mathbf{u}; \mathbf{x} \in \mathbb{Z}^n \times \mathbb{R}^p; x_i = x_t^i, \forall i \notin \mathcal{N}(\mathbf{x}_t)\}, \tag{2}$$

where $x_t^i$ represents the solution of variable $x_i$ at time step $t$. Compared to traditional local search heuristics such as accept ratio and tabu list, LNS is more effective in exploring the solution space, since a larger neighborhood is considered at each step.

## 4 Methodology

In this section, we first model our LNS framework as a MDP. Then, we introduce two important technical components of reinforcement learning agent, i.e., expert policy and agent network. At last, we demonstrate the training procedure.

### 4.1 MDP formulation

We model the LNS problem as a MDP and leverage the techniques of reinforcement learning to automatically learn the optimal strategy. The destroy operator is regarded as the agent and the repair process of solving the sub-MILP problem formulates the environment. At each step, the agent select the subset of destroy variables and environment solves the sub-MILP and provides the reward feedback, which assists the agent in updating its policy. Specifically, we define the key elements of MDP for modeling LNS as follows:

- **State.** The state $\mathbf{s}_t \in S$ at time $t$ includes both the static information from bipartite graph $G$ in a MILP problem and the dynamic information from solution history $\mathbf{x}_{1:t}$.

- **Action.** At each step, the agent selects a subset of variables to be destroyed via taking binary decisions $\mathbf{a}_t$, i.e., $\mathcal{A} = \{0, 1\}^n$. To improve the decision efficiency, we only consider the decisions of discrete variables and keep all continuous variables as destroyed variables since they can be repaired more easily.

- **Transition.** Repair operator solves the MILP sub-problem that only selected variables are optimized and the others are kept unchanged, as shown in Eq.2. The new state $\mathbf{s}_{t+1}$ is determined by updating $\mathbf{s}_t$ with the new solutions $\mathbf{x}_{t+1}$.

- **Reward.** We take the improvement of objective as the reward, i.e., $r_t = r(\mathbf{s}_t, \mathbf{a}_t) = \mathbf{c}^\top(\mathbf{x}_t - \mathbf{x}_{t+1})$. Different problems has different scales that affect the reward. To reduce its impact, we take normalization for $\mathbf{c}$.

The policy $\pi(\mathbf{a}_t|\mathbf{s}_t)$ denotes the probability of selecting a variable in a given state $s_t$. The goal of LNS is to find the optimal policy that maximizes the expected return $\mathbb{E}[R] = \mathbb{E}[\sum_{k=t}^{T} \gamma^{k-t} r_k(\mathbf{s}_t, \mathbf{a}_t)]$, where $\gamma \in [0, 1]$ denotes the discount factor. Directly applying traditional policy-based method may suffer from unstable convergence and low sampling efficiency (Paduraru et al., 2021). To address the issues, we proposes a novel expert guided reinforcement learning framework shown in Fig.1. It consists of two components: the expert and the agent. The expert provides a variable selection guidance for the agent while the agent policy decides the variable selection strategy for the destroy operator.

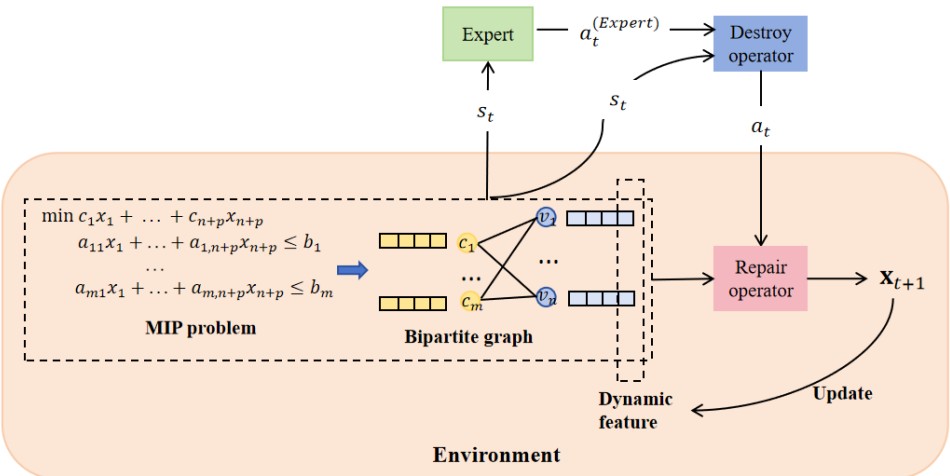

Figure 1: The interaction procedure.

## 4.2 Expert Policy

The purpose of the expert is to provide the variable selection guidance for the agent in online training. The expert policy should be optimized offline via supervised samples from the solver experts. Directly applying solvers to obtain the samples usually has low efficiency, especially when there are a large number of variables and constraints. In this paper, we use Local Branching (LB) (Fischetti & Lodi, 2003) to generate fragment trajectories from the MILP training set for imitation learning. Since generating each sample should call a solver to solve the new MILP problem, which is time consuming. To improve the data generating efficiency, we use the redundant multiple solutions during solving the new problem and thus have multiple expert's decisions $A_t = \{(\mathbf{a}_{t,j}, \mathbf{x}^*_{t+1,j})\}_{j=1}^L$. Thus, we can obtain the experts sample set as $D = \{(\mathbf{s}_t, A_t)\}_{t=1}^T$.

With the expert sample set $D$, we can optimize the expert policy that mimic the solver expert's decisions. If we treat the combination of each variable selection as one class, the class size is exponential and the expert policy is hard to optimize due to the sparsity of each class. Similar to previous work (Wu et al., 2021), we assume that each variable can decide its own decision independently. More specifically, we have:

$$\pi\left(\mathbf{a}_t \mid \mathbf{s}_t\right) = \prod_{i=1}^n \pi^i\left(a_t^i \mid \mathbf{s}_t\right), \tag{3}$$

where $\pi^i\left(a_t^i \mid \mathbf{s}_t\right)$ is the probability of the $i$-th variable being selected under policy $\pi$ at state $\mathbf{s}_t$. In this way, we transform the multi-classification problem into a binary classification problem with $n$ independent variables.

To take full usage of the multiple actions for one certain state, we construct the sample weight for the samples with the same state via energy functions with objective values. The actions with smaller objective value should be assign more sample weight. For an action $\mathbf{a}_i$ with corresponding solution $\mathbf{x}_{i,j}$, a conditional probability can be expressed as

$$w_{i,j} = \frac{\exp\left(-E(\mathbf{x}_{i,j}; \mathbf{s}_i)\right)}{\sum_{j=1}^L \exp(-E(\mathbf{x}_{i,j}; \mathbf{s}_i))}, \tag{4}$$

where $E(\mathbf{x}^{i,j}; \mathbf{s}_i) = \mathbf{c}^i \mathbf{x}_{i,j}$ if $\mathbf{x}^{i,j}$ is a feasible solution and $E(\mathbf{x}^{i,j}; \mathbf{s}_i) = -\infty$ otherwise. With the assistant of the sample weight, we have the loss function of imitation learning:

$$\mathcal{L}(\phi) = -\sum_{i=1}^{|D|} \sum_{j=1}^L w_{i,j} \mathbf{a}_{i,j} \log \pi_\phi(\mathbf{a}_{i,j}; \mathbf{s}_i). \tag{5}$$

### 4.3   Agent Network

For the agent in RL, we take the actor-critic framework to learn the agent's policy. The actor's network generate the variables selecting action and the critic's network takes the value score to assist in evaluating the actor's performance. Both of the actor's network and critic's network take the static information from the bipartite graph $G$ and the dynamic information from solution history. We propose a novel graph transformer structure for extracting the static and dynamic information shared by the actor and critic networks.

#### 4.3.1   Actor's network

As previous mentioned, we model the MIP problem as a bipartite graph, which consist of two types of nodes, i.e., variable nodes and constraint nodes. Traditional methods of learning the representation from the bipartite graph are GCN-based methods that perform two interleaved half-convolutions, one from variables to constraints and one from constraints to variables. Due to its limited receptive field, GCNs struggle to capture this implicit global relations between constraints and variables. To circumvent this issue, we propose a novel graph transformer framework with two global attention units with linear complexity and a graph convolution layer with two interleaved half-convolutions. We design a variable-pair (constraint-pair) attention unit to provide global interactions between arbitrary variable (constraint) node pairs. The interaction between variables and constraints are captured via the two interleaved half-convolutions. The graph transformer framework is shared by the actor and critic networks, shown in Fig.2.

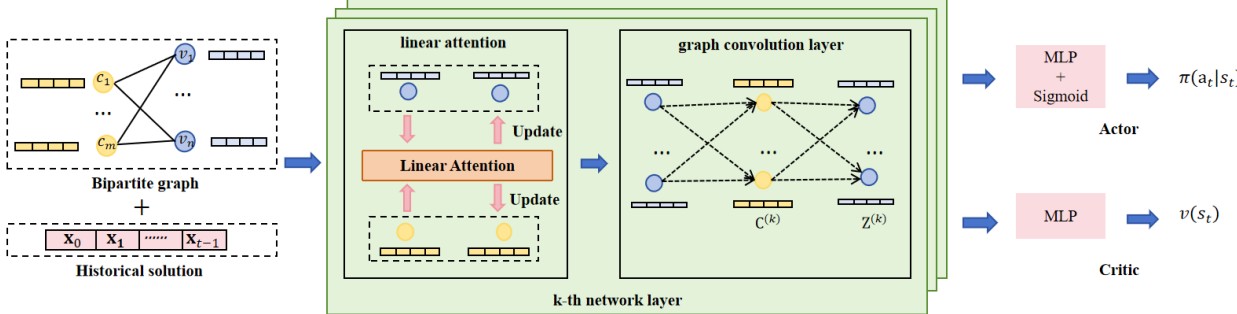

Figure 2: The framework of actor-critic's networks.

Our graph transformer model takes bipartite $\mathcal{G}$ as input to learn the representation. The features of each variable node include the static information in the graph and also the dynamic information including the recent solutions. The features of both the variables and constraint nodes, i.e., $\mathbf{Z}$ and $\mathbf{C}$, are mapped into the latent space, denoted as $\mathbf{Z}^{(0)}$ and $\mathbf{C}^{(0)}$, via a shallow Multi-layer Perception (MLP). Then, we design the variable-pair (constraint-pair) attention units based on a linear attention mechanism (Wu et al., 2023) which can efficiently propagate information among arbitrary nodes with linear complexity. Spatial Encoding (Ying et al., 2021) is used to enable nodes to adaptively attend to all other nodes according to the graph structural information. In specific, for a input embedding $\mathbf{H} \in \mathbb{R}^{N \times d}$, the linear attention function is defined as follows:

$$\tilde{\mathbf{Q}} = \mathrm{norm}\big(f_Q\left(\mathbf{H}\right)\big), \tilde{\mathbf{K}} = \mathrm{norm}\big(f_K\left(\mathbf{H}\right)\big), \tag{6}$$

$$\mathbf{D} = \mathrm{diag}\left(\mathbf{1} + \frac{1}{N}\left(\tilde{\mathbf{Q}}\tilde{\mathbf{K}} + \mathbf{b}\right)\mathbf{1}\right), \tag{7}$$

$$\tilde{\mathbf{H}} \leftarrow \alpha\mathbf{D}^{-1}\left[\mathbf{V} + \frac{1}{N}\left(\tilde{\mathbf{Q}}\tilde{\mathbf{K}}^{\top} + \mathbf{b}\right)\mathbf{V}\right] + (1-\alpha)\mathbf{H}, \tag{8}$$

where $f_Q, f_K, f_V$ are all shallow neural layers, $\mathbf{1}$ is an all-one column vector, the $\mathrm{diag}(\cdot)$ operation changes the vector into a diagonal matrix, the $\mathrm{norm}(\cdot)$ operation takes the vector normalization and and $\alpha$ is a hyper-parameter for residual link. $\mathbf{b}$ is a learnable scalar and $\mathbf{b}_{i,j}$ is indexed by $\psi(v_i, v_j)$, which measures the spatial relation between $v_i$ and $v_j$ in graph $G$. The function $\psi$ can be defined by the connectivity between

the nodes in the graph. In this paper, we choose $\psi(v_i, v_j)$ to be the distance of the shortest path (SPD) between $v_i$ and $v_j$.

The computation complexity above can be efficiently achieved in $\mathcal{O}(N)$. Both the variable-pair attention unit and constraint-pair attention unit adopt the linear attention mechanism but with different parameters. The variable-pair (constraint-pair) attention unit input the variable (constraint) embedding $\mathbf{Z}^{(0)}$ ($\mathbf{C}^{(0)}$) and output $\tilde{\mathbf{Z}}$ ($\tilde{\mathbf{C}}$).

We adopt the graph convolution layer with two interleaved half-convolutions to capture the interaction from the variable node to the constraint node, and the interaction from the constraint node to the variable node. The two interleaved half-convolutions can be expressed as below:

$$\mathbf{c}_i^{(1)} \leftarrow f_C\Big(\tilde{\mathbf{c}}_i, \sum_{j:(i,j)\in\mathcal{E}} g_C\left(\tilde{\mathbf{c}}_i, \tilde{\mathbf{z}}_j, \mathbf{e}_{i,j}\right)\Big), \mathbf{z}_j^{(1)} \leftarrow f_Z\Big(\tilde{\mathbf{z}}_j, \sum_{i:(i,j)\in\mathcal{E}} g_Z\left(\tilde{\mathbf{c}}_i, \tilde{\mathbf{z}}_j, \mathbf{e}_{i,j}\right)\Big), \tag{9}$$

for all $i \in \mathcal{C}$, $j \in \mathcal{V}$, where $f_C, f_Z, g_C, g_Z$ are perceptrons with Tanh activation functions. The output of graph convolution $(\mathbf{Z}^{(1)}, \mathbf{C}^{(1)})$ can further input into the simple transformer units and start a new graph transformer layer. We denote the final output as $(\mathbf{Z}^{(K)}, \mathbf{C}^{(K)})$, where $K$ is the number of graph transformer layer.

Based on the graph transformer layers, we can obtain the state $\mathbf{s}_t$ and design the actor's policy as:

$$\mathbf{s}_t = \mathbf{Z}^{(K)}, \ \pi(\mathbf{a}_t|\mathbf{s}_t) = \text{sigmoid}\big(h_a(\mathbf{s}_t)\big), \tag{10}$$

where $h_a(\cdot)$ is the MLP layers specified for learning the actor's policy. We denote optimized parameters of actor including the parameters of the graph transformer layers and the parameters of $h_a$ as $\theta$.

**Theorem 1.** *The proposed graph transformer can be more expressive than 1-WL Test.*

*Proof.* It is obvious that any GNN can be expressed by our proposed graph transformer since the proposed graph transformer contains the GNN modular. Then, we consider an example by Chen et al. (2023), which is shown in Figure 3, to demonstrate that our graph transformer is more expressive than GNNs.

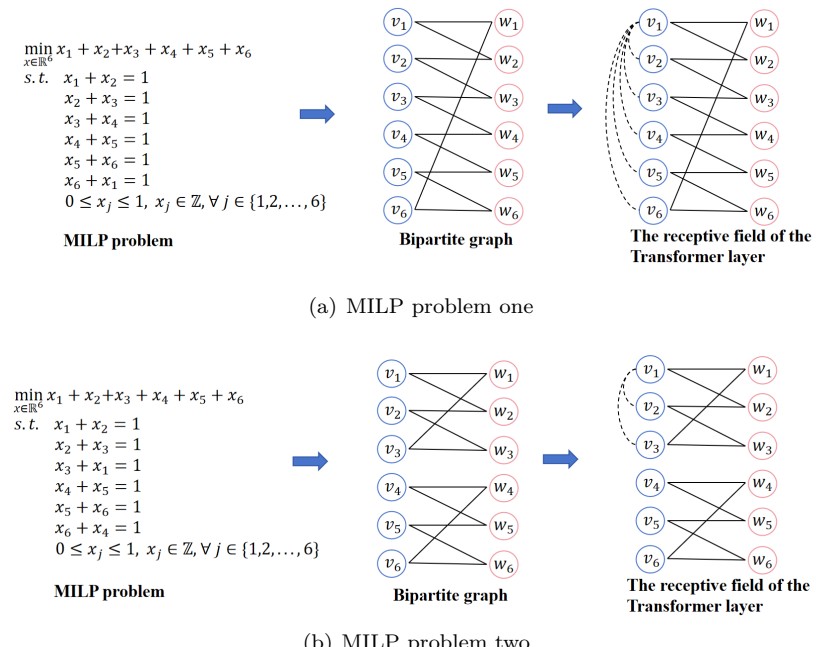

(a) MILP problem one

(b) MILP problem two

Figure 3: An example of non-isomorphic MILP illustration. The dotted line denotes the receptive feld of the Transformer layer.

In each subgraph of Figure 3, the left part is the MILP problem, while the middle one is the corresponding bipartite graph. The right part shows the the receptive field of the transformer layer at node $v_1$. As stated by Chen et al. (2023), the two non-isomorphic MILP graphs cannot be distinguished by WL test. However, since we have the shortest path position encoding, the two graphs have different features even though the two graphs have the same vertex features. When we consider node $v_1$, the shortest paths for two graphs are $\{0, 2, 4, 6, 4, 2\}$ and $\{0, 2, 2, \infty, \infty, \infty\}$. Thus, the transformer can distinguish the two graphs. □

This theorem states the advantage of our graph transformer over GNNs. As stated by Chen et al. (2023) the GNNs should be at most expressive as the 1-WL Test. We give a non-isomorphic MILP illustration in Fig.3, where the GCNs cannot distinguish these two bipartite graphs. However, our graph transformer can have much higher expressive power than 1-WL test as its global receptive field and position encoding. It can distinguish non-isomorphic MILP illustration in Fig.3 successfully. In addition, the position encoding is the relative feature, which represents the degree of correlation via the constraint and can generalize well in the new problems.

### 4.3.2 Critic's network

The critic is designed to estimate the expected return for state $s$, i.e., $V(\mathbf{s}) = \mathbb{E}\left[R_t \mid \mathbf{s}_t = \mathbf{s}\right]$. The critic's network is similar to the actor's network. Both the actor's network and the critic's network share the graph-transformer layers to extract the static representation and the dynamic representation for the solution history. Based on the shared state $s_t$, the critic's network is designed as

$$v(\mathbf{s}_t) = h_c(\mathbf{s}_t), \tag{11}$$

where $h_c(\cdot)$ is the MLP layers specified for learning the critic's value. We denote the critic's parameters including the parameters of graph transformer layers and the parameters of $h_c$ as $\psi$.

### 4.4 Training algorithm

Training the actor-critic network in LNS is a challenge task. As known to all, RL algorithms are unstable during the training. The samples from the MDP in different MILPs follow much different distributions depending on the size and the complexity of MILPs. The diversity of MILPs makes the unstable problem more serious. In addition, to obtain each sample in the MDP, the solver in the environment is called to solve the sub-MILP problem, which is time consuming. Since RL algorithms suffers from low efficiency of learning from samples, the time cost of convergence would be unacceptable. To address these issues, we propose a novel expert guided RL framework based on PPO algorithm.

The PPO algorithm proposed by Schulman et al. (2017) is more stable during the training since it limits the change of parameters and widely used in many RL's applications. In the data collection step of PPO, the old policy network interacts with the environment $T$ times to obtain a complete trajectory $\tau = \{\mathbf{s}_0, \mathbf{a}_0, \mathbf{s}_1, \mathbf{a}_1, \cdots, \mathbf{s}_T, \mathbf{a}_T\}$. Note that the trajectory size $T$ should not be too large as the LNS would not improve any more after only few steps. Based on the trajectory, it computes advantage estimates as

$$\hat{A}_t = \delta_t + (\gamma\lambda)\delta_{t+1} + \cdots + \cdots + (\gamma\lambda)^{T-t+1}\delta_{T-1}, \tag{12}$$

where $\delta_t = r_t + \gamma V_\psi\left(\mathbf{s}_{t+1}\right) - V_\psi\left(\mathbf{s}_t\right)$. Besides the key elements in MDP, we also store the advantage $A_t$ and the value estimation $\Delta_t = r_t + \gamma V_\psi(\mathbf{s}_{t+1})$ for each sample. We sample $M$ MILP instances to generate $M \times T$ samples, from which a mini-batch $\mathcal{B}$ is randomly sampled.

We also assume that each variable can decide its own decision independently as in expert policy. Then, the PPO loss $L^{(\mathrm{A})}(\theta)$ that optimizes the actor's network is:

$$\mathcal{L}^{(\mathrm{A})}(\theta) = \frac{1}{|\mathcal{B}|}\sum_{j\in\mathcal{B}}\left[\frac{1}{n}\sum_{i=1}^{n}\min\left(\left(r_j^i(\theta)\hat{A}_j, C_\epsilon\left(r_j^i(\theta)\right)\hat{A}_j\right)\right)\right], \tag{13}$$

where $r_j^i(\theta) = \frac{\pi_\theta\left(a_j^i|\mathbf{s}_j\right)}{\pi_{\theta_{\mathrm{old}}}\left(a_j^i|\mathbf{s}_j\right)}$, $\epsilon$ is a hyper parameter and $C_\epsilon(r) = \mathrm{clip}(r, 1-\epsilon, 1+\epsilon)$.

To improve the efficiency of learning, we introduce the expert policy to guide the actor's policy during the online training. We should differentiate the guidance of different sample. For the sample with good performance in the old policy, the impact of expert's policy should be reduced. Otherwise, we should improve the guidance of expert. To achieve this goal, we adopt the expert policy learned in the previous subsection and design the weighted guiding loss as:

$$\mathcal{L}^{(\mathrm{E})}(\theta) = \sum_{j \in \mathcal{B}} \left[ \frac{1}{n} \sum_{i=1}^{n} w_j \left( \pi_\phi(a_j^i \mid \mathbf{s}_j) - \pi_\theta^i(a_j^i \mid \mathbf{s}_j) \right)^2 \right], \tag{14}$$

where $w_j = \frac{\exp(-r_j/\tau)}{\sum_{j \in \mathcal{B}} \exp(-r_j/\tau)}$ represents the sample-level weight and $\tau$ is the temperature coefficient. The samples with more objective improvement should be assigned smaller weight with less guidance from the expert. Note that the weight can be also be in instance-level via assigning the total improvement weight during the trajectory as the weight of each sample.

Combining the PPO's actor loss and the expert guiding loss, we can have the total loss that optimize the actor's network as follows:

$$\mathcal{L}(\theta) = \mathcal{L}^{(\mathrm{A})}(\theta) + \alpha \mathcal{L}^{(\mathrm{E})}(\theta), \tag{15}$$

where $\alpha$ balances the actor's loss and the guiding loss.

Different with the actor's network, we do not evaluate each variable independently. Instead, we treat all variables as a whole and have the critic loss as follows:

$$\mathcal{L}(\varphi) = \frac{1}{|\mathcal{B}|} \sum_{j \in \mathcal{B}} \left[ (\Delta_j - V_\psi(\mathbf{s}_j))^2 \right]. \tag{16}$$

Our algorithm for training the actor-critic network for LNS is depicted by the pseudo code in Algorithm 1.

---

**Algorithm 1** Framework of training the actor-critic network for LNS

---

**Input:** actor $\pi_\theta$ with parameters $\theta$; expert $\pi_\phi$; critic $V_\psi$ with parameters $\psi$; empty reply buffer $\mathcal{D}$; step limit $T$; number of iterations $J$; number of epochs $K$; discount factor $\gamma$; $\lambda$ the hyper-prparameter of GAE; learning rates; $\alpha$ the hyper-prparameter to balance the actor's loss and the guiding loss.

**Output:** actor policy $\pi_\theta$.

 1: **for** iteration$= 1, 2, ..., J$ **do**
 2:     draw $M$ MILP instances;
 3:     **for** $m = 1, 2, ..., M$ **do**
 4:         **for** $t = 1, 2, ..., T$ **do**
 5:             sample $\mathbf{a}_t \sim \pi_{\theta_{\mathbf{old}}}(\mathbf{a}_t|\mathbf{s}_t)$;
 6:             receive reward $r_t$ and next state $\mathbf{s}_{t+1}$;
 7:         **end for**
 8:         calculate $\Delta_1, \ldots \Delta_T$ and $\hat{A}_1 \ldots \hat{A}_T$ based on Eq.12;
 9:         store $\{(\mathbf{s}_t, \mathbf{a}_t, \Delta_t, \hat{A}_t)\}$ in $\mathcal{D}$;
10:     **end for**
11:     **for** $k = 1, 2, ..., K$ **do**
12:         randomly sample a mini-batch $\mathcal{B}$ from $\mathcal{D}$;
13:         Optimize actor with parameters $\theta$ based on Eq.15 and critic with parameters $\psi$ based on Eq.16;
14:     **end for**
15:     $\theta_{\mathbf{old}} \leftarrow \theta$;
16: **end for**

---

## 4.5 Searching

LNS is a neighborhood search algorithm operating in the solution space. At each iteration, the destroy operator perturbs the current solution to define a neighborhood, and the repair operator then searches

that neighborhood for improvements. In our approach, the destroy operator is the actor network trained via reinforcement learning. Specifically, given a MILP instance and its current solution, convert it into a bipartite graph and feed it into the actor network, which outputs the destruction probability for each variable. Only the $k$ variables with the highest probabilities are selected and optimized. If the repair operator finds a better solution, it replaces the current one. The search process is shown in Algorithm 2.

---

**Algorithm 2** Searching algorithm

---

**Input:** actor $\pi_\theta$ with parameters $\theta$; neighbor radius $k$; step limit $T$; a MILP instance; initial feasible solution $\mathbf{x}_0$.

**Output:** $\mathbf{x}_{T+1}$.

1: **for** $t = 0, 1, ..., T$ **do**
2:     Obtain the state $\mathbf{s}_t \in S$ at time $t$;
3:     Select the $k$ variables with the highest probabilities based on $\pi_\theta(\mathbf{a}_t|\mathbf{s}_t)$ to obtain the sub-problem;
4:     Solve the sub-MILP problem to obtain a new feasible solution $\mathbf{x}'$ that minimizes the objective function value;
5:     **if** $\mathbf{c}^\top \mathbf{x}' < \mathbf{c}^\top \mathbf{x}_t$ **then**
6:         $\mathbf{x}_{t+1} \leftarrow \mathbf{x}'$;
7:     **else**
8:         $\mathbf{x}_{t+1} \leftarrow \mathbf{x}_t$;
9:     **end if**
10: **end for**

---

## 5 Experiments

In this section, we perform experiments to evaluate the performance of our method. We first introduce experimental settings, including the dataset, features, the hyper-parameters and baselines in our experiments. Then, we provide an empirical evaluation to demonstrate the effectiveness of our proposed algorithm. At last, we provide an in-depth analysis from the convergence and the network structure.

### 5.1 Experiment Settings

#### 5.1.1 Benchmarks

We perform experiments on three NP-hard benchmark problems: Set Covering (SC), Combinatorial Auction (CA) and Balanced Item Placement (IP). The details of the problems are in the Appendix A. We generate SC instances with 300 columns and 400 rows for training, following the procedure by Balas & Ho (1980). Instances in test set are generated with 400 columns and 500 rows. CA instances are generated based on Leyton-Brown et al. (2000), with 400 items and 300 bids in training and valid set, 500 items and 400 bids in test set. The IP instances, with 78 constraints, 15 continuous variables and 168 binary variables, come from the NeurIPS ML4CO 2021 competition (Gasse et al., 2022).

We selected these benchmarks because they are particularly challenging for state-of-the-art solvers and reflect the types of MILP problems commonly encountered in practical applications. SC encapsulates the essence of MILP since column generation formulations can be applied to virtually any complex discrete optimization problem. The distributions in CA are economically motivated and model real-world problems. Additionally, IP is a classic combinatorial optimization problem that involves both discrete and continuous variables, making it difficult to solve even with a limited number of constraints and variables.

### 5.2 Baselines

We compare with five baselines: (1) Optimal, the best solution obtained by Gurobi; (2) RAND, a LNS method with the policy of randomly selecting the neighborhoods; (3) IL-LNS (Sonnerat et al., 2022), a supervised learning algorithm by imitating Local Branching; (4) CL-LNS (Huang et al., 2023), a contrastive learning

method that collects positive and negative solution samples from an expert under the LNS framework; (5) R-LNS (Song et al., 2020), a LNS method that randomly groups variables into disjoint subsets of equal size and re-optimize them in order.

### 5.2.1 Features

To solve the MILP problems, we propose a set of features to describe the current state of the problem. This set of features include 7 variable features, 4 constraint features and 1 edge features. Among them, the solution value at the current step is dynamic, while others are all static. All the features can be easily obtained either directly from the original MILP problem or by conducting simple calculations. The details of these features are in the Appendix B.

### 5.2.2 Metrics

To compare solution quality, we use the following metrics: (1) The *primal bound* is the objective value of the MILP; (2) The *objective improvement* is defined as $\left| \mathbf{c}^\top \tilde{\mathbf{x}}_T - \mathbf{c}^\top \mathbf{x}_0 \right|$ to show the improvement of the MILP; (3) The *primal gap* (Khalil et al., 2017) is defined as $\left| \mathbf{c}^\top \mathbf{x}_T^* - \mathbf{c}^\top \tilde{\mathbf{x}}_T \right| / \left| \mathbf{c}^\top \tilde{\mathbf{x}}_T \right| \cdot 100\%$ for each instance to show the gap from the best solution found by all methods; (4) The *primal integral* (Achterberg et al., 2012) at time $q$ is the integral on $[0, q]$ of the primal gap as a function of runtime. It captures the quality of and the speed at which solutions are found;

### 5.2.3 Hyper-parameters

We conduct experiments on 2.60 GHz Intel(R) Xeon(R) Gold 6240 CPU. Trainings are done on a NVIDIA A100 GPU with 40 GB memory. The training set, validation set, and test set have 300, 100, and 30 instances respectively. We use the state-of-the-art IP solver Gurobi (v11.0.0) as the repair operator to solve the sub-problem at each step of LNS with time limit for 5 seconds for SC and CA, 10 seconds for IP. The initial solution of each instance is also given by Gurobi. The destroy degree of all approaches is set to 0.1 except for R-LNS, in which the neighborhood size is defined implicitly by the the policy it used. We set the training and testing step limit $T = 4$. For RL, we use $\epsilon = 0.2$ for probability clipping and $\gamma = 0.98$, $\lambda = 0.95$, $\alpha = 100$. We use Adam optimizer with learning rate $1 \times 10^{-4}$ for both policy and value network. For each problem, we train 600 iterations, during each we draw $M = 32$ instances as a batch. The length of the experience replay is $TM$, the number of updating the network is $U = 4$ and the mini-batch size is $\mathcal{B} = TM/U$. The shortest path length between nodes is divided into 2, 4, 6 or more for position encoding. We applied full attention once to variables and constraints respectively, and then used two half-convolution layers. For IL-LNS, we use LB to collect data with the step limit 4 and the destroy degree 0.1. We collect 1147,1068 and 1199 state-action pairs for SC, CA and IP respectively. We train 1000 epochs with the batch size 32, using Adam optimizer with learning rate $1 \times 10^{-3}$. We divide the training set and valid set in a ratio of 1:9. For CL-LNS, we use the parameters as Huang et al. (2023), but employed our features proposed above. We train 100 epochs for each problem with learning rate $1 \times 10^{-3}$ and batch size 32.

## 5.3 Results

### 5.3.1 Overall Comparison.

In our experiments, we observed issues with overfitting and instability in the learning based methods. To address this, we conducted tests every 100 training epochs and selected the best result as the final outcome. The results are gathered in Table 1.

The experimental results indicate that, under the same step size and neighborhood radius, our method significantly outperforms all baselines across all problem sets, yielding solutions that closely approximate the optimal ones. Given the practical need for fast, high-quality solutions to IP problems, our method, which operates with a low neighborhood radius and fewer LNS iterations, shows great potential for solving difficult problems.

Table 1: Comparison with LNS baselines.

| Methods | SC | | | | CA | | | | IP | | | |
|---|---|---|---|---|---|---|---|---|---|---|---|---|
| | PB | Imp. | PG% | PI | PB | Imp. | PG% | PI | PB | Imp. | PG% | PI |
| Optimal | 1031.00 | 4392.07 | 0.00 | - | -20766.55 | 5782.47 | 0.00 | - | 8.37 | 204.24 | 0.00 | - |
| RANDOM | 4494.89 | 418.18 | 77.06 | 34.60 | -15074.36 | 90.28 | 37.76 | 20.23 | 142.86 | 69.74 | 94.14 | 94.62 |
| IL-LNS | 1351.20 | 4061.87 | 23.70 | 15.53 | -18229.39 | 3245.31 | 13.92 | 6.76 | 20.75 | 191.85 | 59.69 | 83.65 |
| CL-LNS | 3719.03 | 1694.03 | 72.28 | 21.78 | -15504.44 | 520.36 | 33.94 | 13.44 | 101.94 | 110.66 | 91.79 | 76.31 |
| R-LNS | 2257.77 | 3155.30 | 54.30 | 42.01 | -15846.32 | 880.24 | 31.05 | 16.22 | 114.20 | 98.40 | 92.68 | 70.09 |
| **Ours** | **1121.77** | **4291.30** | **8.09** | **12.88** | **-18283.01** | **3298.93** | **13.58** | **6.33** | **16.59** | **196.02** | **49.57** | **65.75** |

### 5.3.2  Ablation Study.

To validate the effectiveness of proposed modules, we individually remove the applied techniques in the two major parts of our method, i.e., the RL and the Transformer module shown in Table 2.

Table 2: Comparison with variants of our method.

| Methods | SC | | | | CA | | | | IP | | | |
|---|---|---|---|---|---|---|---|---|---|---|---|---|
| | PB | Imp. | PG% | PI | PB | Imp. | PG% | PI | PB | Imp. | PG% | PI |
| Optimal | 1031.00 | 4392.07 | 0.00 | - | -20766.55 | 5782.47 | 0.00 | - | 8.37 | 204.24 | 0.00 | - |
| w/o RL | 1274.30 | 4138.77 | 19.09 | 15.07 | -18013.17 | 3029.09 | 15.29 | 6.67 | 19.65 | 192.95 | 57.40 | 69.20 |
| w/o Trans | 1154.20 | 4258.87 | 10.67 | 13.46 | -18241.20 | 3257.12 | 13.84 | 6.36 | 18.94 | 193.66 | 55.81 | 83.02 |
| w/o RL&Trans | 1351.20 | 4061.87 | 23.70 | 15.53 | -18229.39 | 3245.31 | 13.92 | 6.76 | 20.75 | 191.85 | 59.66 | 83.65 |
| **Ours** | **1121.77** | **4291.30** | **8.09** | **12.88** | **-18283.01** | **3298.93** | **13.58** | **6.33** | **16.59** | **196.02** | **49.57** | **65.75** |

We draw the following major conclusions: Firstly, when the Transformer module is removed (w/o Trans), there is a noticeable performance drop across all datasets, highlighting its broad receptive field that effectively captures relationships between variables or constraints efficiently. The combined operation of these layers is crucial for solving MILP problems and achieving a better understanding of the problem model. Secondly, the exclusion of reinforcement learning (w/o RL), relying solely on imitation learning, results in compromised outcomes. This suggests that imitation learning alone is limited by the quality of expert data, whereas reinforcement learning enhances the model's understanding and adaptability by interacting with the environment. Furthermore, our findings indicate that even if the expert network guiding the training isn't the optimal network structure, it still achieves results that surpass the best imitation learning outcomes. This demonstrates the immense potential of reinforcement learning in solving MILP problems and highlights the significant room for improvement when combining reinforcement learning with imitation learning methods.

### 5.4  In-depth Analysis

### 5.4.1  Convergence analysis.

We collected data on the total rewards, PPO actor loss, the expert guiding loss, and value network loss for each iteration throughout the training period to demonstrate the convergence of our algorithm. The training status is shown in Figure 4.

The results indicate that the total reward value, expert loss, PPO loss, and value network loss stabilize at fixed values in the later stages of training, confirming the convergence of the model.

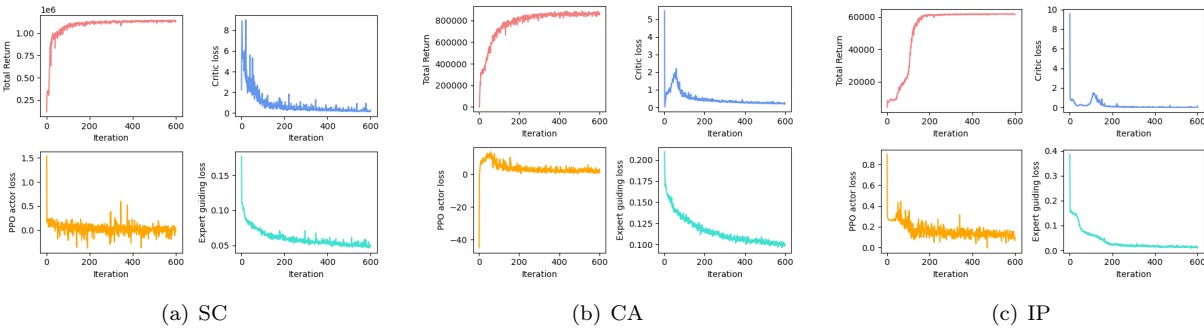

(a) SC               (b) CA               (c) IP

Figure 4: RL training status.

### 5.4.2   Graph-transformer analysis

We tested the effects of transformer and position encoding using four network structures: Graph transformer with position encoding, Graph transformer without position encoding, Half graph transformer with global respective field only on variable side, and GCNs only. Position encoding is only used on transformers for variables. We use the SC as the benchmark problem, which captures the quintessence of integer linear programming, since column generation formulations can be written for virtually any difficult discrete optimization problem. We train on instances with 500 (Easy), 1000 (Medium) and 1500 (Hard) nodes using imitation learning. The size of the testing instances is slightly larger than that of the training instances. The experimental results are shown in Table 3.

Table 3: The effects of graph transformer structure and position encoding.

| Methods | Easy (Vars 500, Cons 600) | | | | Medium (Vars 1000, Cons 1100) | | | | Hard (Vars 1500, Cons 1600) | | | |
|---|---|---|---|---|---|---|---|---|---|---|---|---|
| | PB | Imp. | PG% | PI | PB | Imp. | PG% | PI | PB | Imp. | PG% | PI |
| Graph Trans. + PE | 1393.80 | 4345.63 | 0.00 | 27.06 | 657.80 | 6074.47 | 0.00 | 39.24 | 452.17 | 6887.53 | 0.00 | 40.64 |
| Graph Trans. | 1574.50 | 4164.93 | 11.48 | 25.71 | 695.70 | 6036.57 | 5.45 | 38.68 | 455.80 | 6883.90 | 0.80 | 42.10 |
| Half Graph Trans. | 1607.47 | 4131.97 | 13.29 | 32.03 | 687.90 | 6044.37 | 4.38 | 38.85 | 457.43 | 6882.27 | 1.15 | 41.54 |
| Two GCNs | 1635.17 | 4104.27 | 14.76 | 26.08 | 707.23 | 6025.03 | 6.99 | 39.17 | 470.40 | 6869.30 | 3.88 | 43.77 |

Since LNS are anytime algorithms, we record the time and current objective function value after each LNS step. The destroy degree is set to 5% and we regard the solution given by Gurobi after 20 seconds of solving each problem as the best solution. The algorithm stops when the objective function value does not improve after three steps of LNS. Figure 5 show the primal gap as a function of runtime.

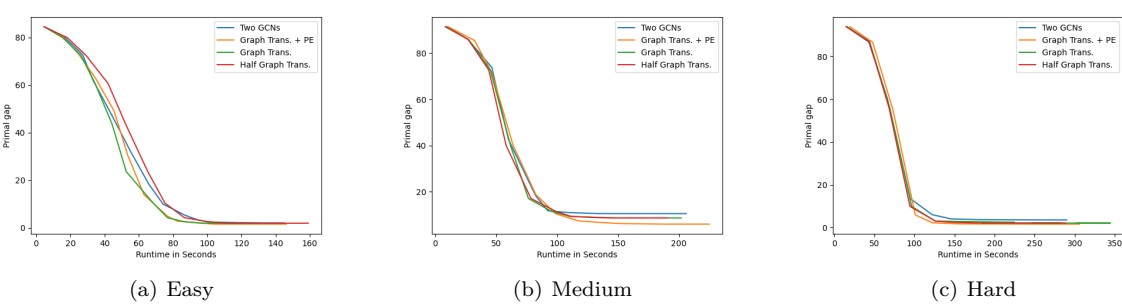

(a) Easy            (b) Medium           (c) Hard

Figure 5: The primal gap of Easy, Medium and Hard instances.

For both the bipartite graphs and position encodings are precomputed, they are not included in the running time. In fact, the bipartite graph for 300 SC problems, each with 1,500 variables and 1,600 constraints, can be generated in just a few tens of seconds. Calculating the position encoding is more time-consuming. However, it can be pre-calculated before training begins and only needs to be computed once. This takes approximately 40 minutes by using the networkx library and Dijkstra's algorithm. In fact, for sparse graphs, there are currently faster algorithms to calculate the shortest path for all pairs and we can also use threshold pruning to reduce the time.

Take the SC problem with 1500 variables and 1600 constraints as an example. With a 5% destroy degree, solving the above problems using only GCNs requires 10 steps of LNS to reach a local optimum, and 30 problems can be solved in about 260 seconds. While using tranformers, they can be solved in 290 seconds and only 8 steps of LNS are required.

The results indicate that all network configurations incorporating global interactions outperform those relying solely on GCNs, highlighting the effectiveness of transformers. Adding position encoding to the graph transformer can produce better training performance, which demonstrates its effectiveness in capturing spatial information. In addition, our model has good performance on problems of different scales, indicating its good generalization ability.

### 5.4.3 Generalization analysis

We analyze the generalization of the model on instances of different sizes. We test on instances of the same size as the training instances as well as test on larger instances to evaluate generalization to larger problems. We select a step size limit of 15. The time limit for Gurobi to solve the sub-problems in each step is 5 seconds for SC and CA, and for 10 seconds for IP. Table 4 ,Table 5 and Table 6 show the experimental results at instances of the same, twice and triple size, respectively. Figure 6, Figure 7 and Figure 8 show the primal gap as a function of runtime at instances of the same, twice and triple size, respectively.

Table 4: Test on instances of the same size as the training instances.

| Methods | SC | | | | CA | | | | IP | | | |
|---|---|---|---|---|---|---|---|---|---|---|---|---|
| | PB | Imp. | PG% | PI | PB | Imp. | PG% | PI | PB | Imp. | PG% | PI |
| Optimal | 1232.83 | 3928.67 | 0.00 | - | -16139.97 | 4592.32 | 0.00 | - | 8.37 | 204.24 | 0.00 | - |
| w/o RL | 1445.50 | 3816.00 | 7.79 | 71.14 | -14200.40 | 2652.76 | 13.65 | 43.22 | 19.50 | 193.09 | 57.12 | 145.94 |
| w/o Trans | 2256.46 | 3005.03 | 40.93 | 164.06 | -14255.03 | 2707.39 | 13.22 | 32.98 | 18.88 | 193.72 | 55.70 | 171.96 |
| w/o RL&Trans | 2288.47 | 2973.03 | 41.75 | 172.77 | -13548.33 | 2000.70 | 19.12 | 36.04 | 19.79 | 192.80 | 57.75 | 174.67 |
| **Ours** | **1409.97** | **3851.53** | **5.47** | **53.59** | **-14384.01** | **2836.37** | **12.20** | **27.90** | **16.58** | **196.01** | **49.56** | **131.68** |

Table 5: Test on instances of twice size as the training instances.

| Methods | SC | | | | CA | | | | IP | | | |
|---|---|---|---|---|---|---|---|---|---|---|---|---|
| | PB | Imp. | PG% | PI | PB | Imp. | PG% | PI | PB | Imp. | PG% | PI |
| Optimal | 864.77 | 5334.40 | 0.00 | - | -31841.82 | 8781.69 | 0.00 | - | 22.258 | 289.95 | 0.00 | - |
| w/o RL | 1115.90 | 5083.27 | 22.51 | 219.81 | -28233.32 | 5173.20 | 12.78 | 5.54 | **38.86** | **251.83** | **98.09** | **218.46** |
| w/o Trans | 1161.33 | 5037.83 | 25.54 | 271.10 | -28088.26 | 5028.13 | 13.36 | 5.09 | 171.88 | 118.81 | 99.57 | 228.86 |
| w/o RL&Trans | 1380.73 | 4818.43 | 37.37 | 333.61 | -27520.13 | 4460.00 | 15.70 | 5.13 | 51.43 | 239.26 | 98.56 | 243.56 |
| **Ours** | **963.03** | **5236.13** | **10.20** | **130.11** | **-28296.26** | **5236.13** | **12.53** | **4.11** | 55.69 | 235.00 | 98.67 | 236.24 |

The results show that the combination of reinforcement learning with Transformer outperforms other models on the SC and CA problems, highlighting the generalization potential of our approach on large-scale problems. However, while our method also achieves the best performance on IP problems of the same size, its generalization ability on larger instances is not as strong as that of imitation learning with Transformer.

Table 6: Test on instances of triple size as the training instances.

| Methods | SC | | | | CA | | | | IP | | | |
|---|---|---|---|---|---|---|---|---|---|---|---|---|
| | PB | Imp. | PG% | PI | PB | Imp. | PG% | PI | PB | Imp. | PG% | PI |
| Optimal | 690.67 | 6103.53 | 0.00 | - | -48101.66 | 13481.20 | 0.00 | - | 3.64 | 342.56 | 0.00 | - |
| w/o RL | 1036.23 | 5757.96 | 33.34 | 31.61 | -39162.17 | 4541.71 | 22.82 | **19.13** | **9.87** | **336.31** | **63.19** | 454.05 |
| w/o Trans | 956.06 | 5838.13 | 27.76 | 26.31 | -37535.70 | 2915.23 | 28.15 | 22.87 | 286.71 | 59.48 | 98.73 | **431.95** |
| w/o RL&Trans | 1049.33 | 5744.86 | 34.18 | 25.49 | -36764.74 | 2144.28 | 30.83 | 21.40 | 65.20 | 280.99 | 94.42 | 446.42 |
| **Ours** | **848.20** | **5946.00** | **18.57** | **16.75** | **-40022.46** | **5402.00** | **12.20** | 19.67 | 189.58 | 156.61 | 98.08 | 526.41 |

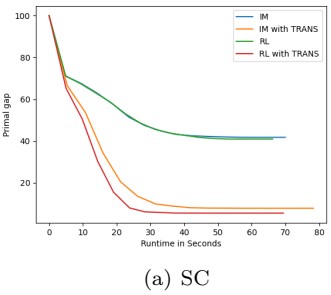
(a) SC

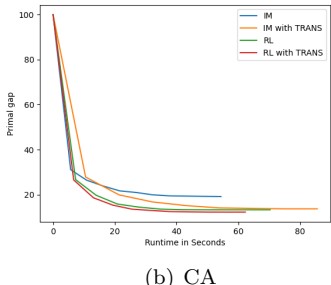
(b) CA

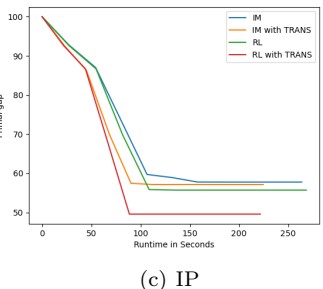
(c) IP

Figure 6: Test results at instances of the same size.

This may be due to the problem generation process: for SC and CA, only a few parameters are needed to generate instances, so increasing the instance size does not significantly alter their structural characteristics. In contrast, generating IP instances requires specifying a large number of parameters, leading to substantial structural changes as the scale increases. These changes can undermine the effectiveness of reinforcement learning. Nevertheless, it is worth noting that Transformer-based methods consistently outperform GCNs, indicating that the Transformer architecture helps mitigate the structural variability challenges in large instances.

### 5.4.4 Efficiency analysis

We analyze the inference efficiency of the proposed model. Table 7 shows the average inference time and difference from the optimal inference time of the proposed method compared to other baselines.

Table 7: Inference time of the proposed method compared to other baselines

| Methods | SC | | CA | | IP | |
|---|---|---|---|---|---|---|
| | Time(s) | Difference(s) | Time(s) | Difference(s) | Time(s) | Difference(s) |
| RANDOM | 21.28 | 13.76 | 13.11 | 5.71 | 24.67 | 4.03 |
| IL-LNS | 7.93 | 0.41 | 7.70 | 0 | 26.66 | 6.02 |
| CL-LNS | 7.52 | 0 | 9.87 | 2.17 | 20.77 | 0 |
| R-LNS | 20.95 | 13.43 | 16.52 | 8.82 | 24.92 | 4.28 |
| **Ours** | 8.56 | 1.04 | 7.85 | 0.15 | 22.17 | 1.53 |

The results show that although our model used Transformer, the inference time does not significantly increase. In addition, due to our model's ability to obtain better solutions compared to other baselines, it has higher efficiency and potential.

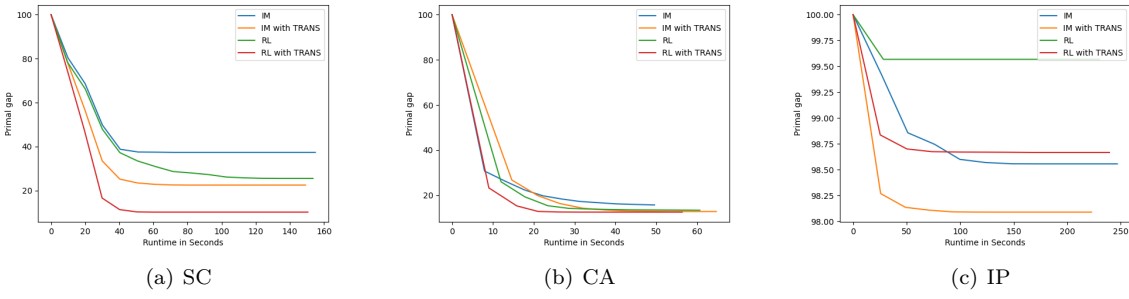

Figure 7: Test results at instances of the twice size.

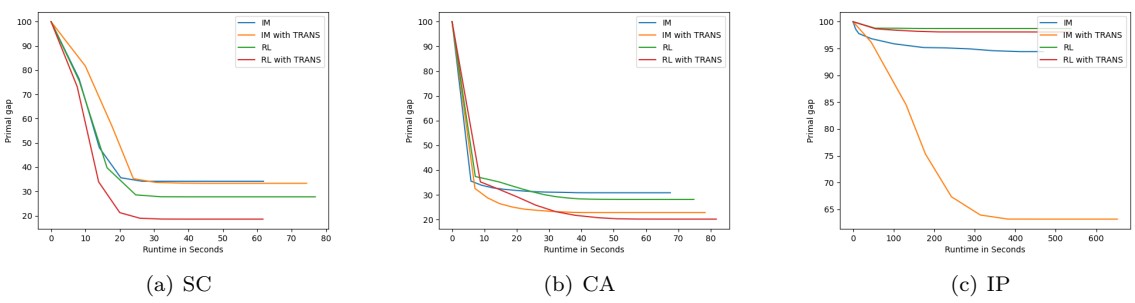

Figure 8: Test results at instances of the triple size.

## 6 Conclusion

In this paper, we proposed a novel expert-guided reinforcement learning model to efficiently solve MILP problems within the LNS framework. We first introduced a graph transformer framework featuring two global attention units with linear complexity and a graph convolution layer employing two interleaved half-convolutions to capture the global relationships among variables and constraints. We then developed a stable actor-critic framework that separates the actor's policies for each variable while evaluating all variables using a critic network. Additionally, we incorporated an expert that provides online weighted guidance during training, based on the PPO algorithm, to enhance learning efficiency. We would explore parallel computing of our method to larger-scale problems as future work.

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

## A  Additional details of instance generation

Experiments are performed on three NP-hard benchmark problems.

**Set Covering (SC)**: For an instance of SC, we have a set $S = \{e_1, e_2, \cdots, e_n\}$. $S_1, S_2, \cdots, S_m$ is a subset of $S$ and $\cup S_j = S$. The objective is to select the smallest possible number of subsets from $S$ such that the union of these subsets still covers the entire universe of elements:

$$\min \sum_{j=1}^{m} x_j \tag{17}$$

$$\text{s.t.} \sum_{j:e_i \in S_j} x_j \geqslant 1, i = 1, 2, \cdots, n \tag{18}$$

$$x_j = \{0, 1\}, j = 1, 2, \cdots, m \tag{19}$$

**Combinatorial Auction (CA)**: For an instance of CA, we have $n$ bids for $m$ items. For each bid, $B_i$ is a subset of items and $p_i$ is its associated bidding price. The objective is to allocate items to bids in order to maximize the total revenue:

$$\min - \sum_{i=1}^{n} p_i x_i \tag{20}$$

$$\text{s.t.} \sum_{i:j \in B_i} x_i \leq 1, j = 1, 2, \cdots, m \tag{21}$$

$$x_i \in \{0, 1\}, i \in 1, 2, \cdots, n \tag{22}$$

**Balanced Item Placement (IP)**: For an instance of IP, let $I$ denote the set of items and $J$ denote the set of containers. Each item will be placed in exactly a single container. Let $K$ represent the set of dimensions. For dimension $k \in K$ of container $j \in J$, knapsack constraints equation 25 represent some physical considerations while equation 26 and equation 27 properly account for the placement unevenness, which is penalized in the objective.

$$\min_{x,y,z} \sum_{j \in J} \sum_{k \in K} \alpha_k y_{jk} + \sum_{k \in K} \beta_k z_k \tag{23}$$

$$\text{s.t.} \sum_{j \in J} x_{ij} = 1, \forall i \in I \tag{24}$$

$$\sum_{i \in I} a_{ik} x_{ij} \leq b_k, \forall j \in J, \forall k \in K \tag{25}$$

$$\sum_{i \in I} d_{ik} x_{ij} + y_{jk} \geq 1, \forall j \in J, \forall k \in K \tag{26}$$

$$y_{jk} \leq z_k, \forall j \in J, \forall k \in K \tag{27}$$

$$x_{ij} \in \{0, 1\}, \forall i \in I, \forall j \in J \tag{28}$$

$$y_{jk} \geq 0, \forall j \in J, \forall k \in K \tag{29}$$

## B  State features

In this paper, we describe the current state of the instance by a bipartite graph, attached by the features of variables, constraints and edges. These features, which are crucial for solving MILP problems, should reflect instance information and solution states. The former is called static features and the latter is called dynamic features. Restricted by time and space consumption, for static features, we pre-calculate and store them separately before training. For dynamic features, considering the efficiency of the solving process, we directly record the current solution for each step of LNS. Link dynamic features with static features of variables and

attach them to variable node as node features. Table 8 shows the details of these features we use in our experiments, including 7 variable features, 4 constraint features and 1 edge features. Only the solution value at the current step is dynamic, while the others are all static.

Table 8: The list of features for variables, constraints and edges.

| Feature Types | Description | Static |
|---|---|---|
| **Variable features (Z)** | Normalized coefficient of variables in the objective function | Yes |
| | Average coefficient of the variable in all constraints | Yes |
| | Maximum value among all coefficients of the variable | Yes |
| | Minimum value among all coefficients of the variable | Yes |
| | Binary representation to show whether the variable is integer variable | Yes |
| | Solution value at the current step. | No |
| **Constraint features (C**) | Average of all non-zero coefficients in the constraint | Yes |
| | Number of all non-zero coefficients in the constraint | Yes |
| | Right-hand-side value of the constraint | Yes |
| | The sense of the constraint | Yes |
| **Edge features (E)** | Coefficient of variables in constraints | Yes |

## C   Experimental details of baselines

In this section, we will show some experimental results of IL-LNS and CL-LNS. Figure 9 show the IL-LNS loss of SC, CA and IP respectively. Figure 10 shows the CL-LNS loss of SC, CA and IP.

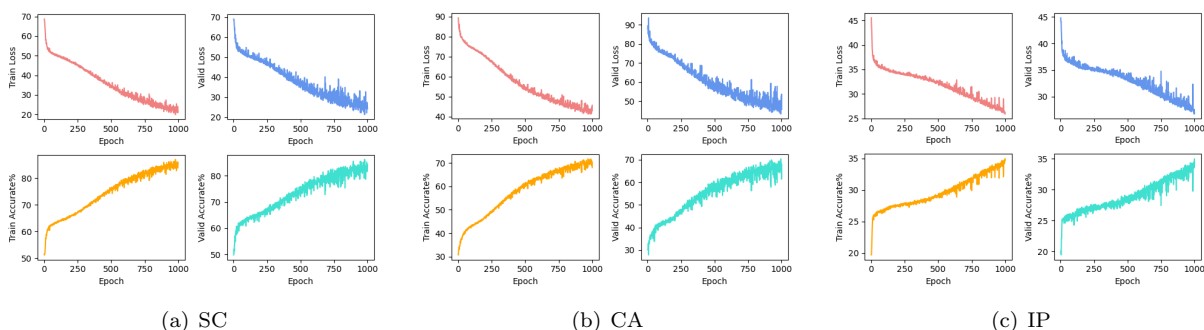

(a) SC                                    (b) CA                                    (c) IP

Figure 9: IL training status.

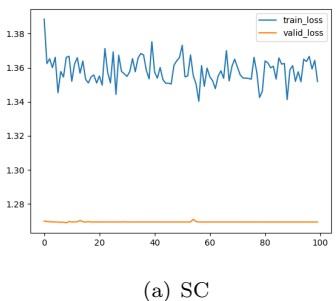
(a) SC

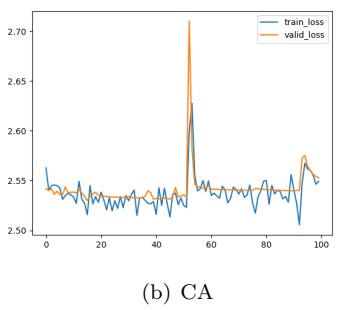
(b) CA

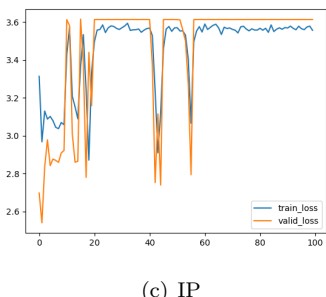
(c) IP

Figure 10: CL training status

# D Influence of different network structures

In this section, we add several different combinations of attention and GCNs to show the effects of different combinations. We use imitation learning and test it on an IP problem instances with 99 discrete variables, 12 continuous variables, and 60 constraints. Table 9 shows the influence of different network structures and from left to right shows the order of each component, where Var (Cons) means global interactions between arbitrary variable (constraint) node pairs, V-C (C-V) means half-convolutions from variable to (constraint) nodes to constraint (variable) nodes.

Table 9: The influence of different network structures.

| Var | Cons | V-C | Cons | C-V | Var | Cons | V-C | Cons | C-V | dropout | PB | PG% |
|-----|------|-----|------|-----|-----|------|-----|------|-----|---------|------|------|
|     |      | Y.  |      | Y.  |     |      | Y.  |      | Y.  |         | 26.98 | 47.81 |
| Y.  | Y.   | Y.  |      | Y.  |     |      | Y.  |      | Y.  |         | 14.28 | 1.40 |
| Y.  | Y.   | Y.  |      | Y.  |     |      | Y.  |      | Y.  | Y.      | 16.10 | 12.55 |
| Y.  |      | Y.  | Y.   | Y.  | Y.  |      | Y.  | Y.   | Y.  |         | 16.47 | 14.51 |
| Y.  |      | Y.  | Y.   | Y.  | Y.  |      | Y.  | Y.   | Y.  | Y.      | 17.26 | 18.42 |
| Y.  | Y.   | Y.  |      | Y.  | Y.  | Y.   | Y.  |      | Y.  |         | 14.08 | 0.00 |
| Y.  | Y.   | Y.  |      | Y.  | Y.  | Y.   | Y.  |      | Y.  | Y.      | 42.11 | 66.56 |

The results indicate that all network configurations incorporating global interactions outperform those relying solely on GCNs, highlighting the effectiveness of transformers in this context. Furthermore, we find that combining global interactions between arbitrary variable node pairs and constraint node pairs as a unified approach, along with using two half-convolutions as a cohesive strategy, leads to better training performance and more effectively captures the model's structure.

However, using only a single-layer transformer for variables and constraints in the first layer is insufficient to fully capture global relationships. By adding an extra transformer layer on top of GCNs, we can further integrate variables and constraints from a global perspective, thereby enhancing the capture of these relationships.

Additionally, our findings suggest that using dropout in attention mechanisms may result in performance degradation and slower convergence. Therefore, we recommend avoiding dropout when employing transformers to solve MILP problems.

