# OpenReview forum: "Graph Transformer based Large Neighborhood Search via Expert Guided Reinforcement Learning"
_TMLR — Rejected by TMLR_

### Review · Reviewer_66od · 2025-04-27

**Summary Of Contributions:**

This paper enhances the LNS studies in solving MILP. It has two main improvements: 1) adding the Transformer module to GNN, aiming to improve the expression ability; 2) incorporating the imitation loss into the actor training of RL. The method is tested on three problems: Set Covering (SC), Combinatorial Auction (CA), and Balanced Item Placement (IP).

**Audience:**

Yes

**Claims And Evidence:**

No

**Requested Changes:**

See weaknesses. I believe this paper is not ready for acceptance.

**Strengths And Weaknesses:**

## Strengths:
1. The code is provided.
2. The idea of incorporating the imitation loss in RL is interesting.

---

## Weaknesses:
1. The proof that the graph transformer can be more expressive than the 1-WL Test is not general. Whether this expressiveness can generally enhance performance remains unproven and empirically unverified. For example, the results on CA in Table 2 show that adding the Transformer module leads to worse performance when using only imitation learning.
2. The illustration in Figure 3 and the corresponding motivation are very similar to [1]. The paper lacks a comparison with [1].
3. Why is the primal gap of CL-LNS so different from the original paper? For example, the original CL-LNS paper shows a primal gap of less than 1%, but it is 33% and 72% in this paper on the same problems — CA and SC.
4. The metrics used in this paper are uncommon in this field, where primal gap and primal integral are typically reported. What does objective improvement mean? Why are they not zero for optimal solutions?
5. The paper lacks a comparison with Wu et al., 2021, which also uses RL to train the LNS.
6. What is the inference time of the proposed method compared to other baselines? How many the improvement steps are applied during inference?
7. What is the impact of the weight introduced in Equation (5)? How to balance the RL loss and the imitation loss?
8. The imitation targets obtained from local branching introduce additional computational overhead.
9. The generalization performance of the method is unclear. Would be better to test it on benchmark datasets like Distributional MIPLIB (https://sites.google.com/usc.edu/distributional-miplib), which includes MILP instances with different distributions and levels of hardness.


[1] Ye, Xinyu, et al. "On Designing General and Expressive Quantum Graph Neural Networks with Applications to MILP Instance Representation." The Thirteenth International Conference on Learning Representations. 2025.

---

> ### Author Response · Authors · 2025-06-06
> **Response to Reviewer 66od (part 1 of 2)**
>
> Thanks for the detailed reviews, and we apologize for the confusion. We would like to address with the followings.
> * **Q1**: The proof that the graph transformer can be more expressive than the 1-WL Test is not general. Whether this expressiveness can generally enhance performance remains unproven and empirically unverified. For example, the results on CA in Table 2 show that adding the Transformer module leads to worse performance when using only imitation learning.
>
> * * **A1**: Thank you for the kind reminder. In fact, Theorem 1 and its proof is a general result. It tells that the graph transformer can deal with non-isomorphic MILP graphs, but the GCN cannot. However, it does not mean that more expressive ability should have better experiment result at all datasets and algorithms. These two concepts are not equal. When a dataset contains only isomorphic MILP graphs, the graph transformer no longer has an expressive advantage. Moreover, the graph transformer contains more parameters and may be overfitting easily when the dataset is not large enough. This problem is more obvious in the imitation learning since it relies on the local branching method to generate the samples offline while RL would generate more samples online. Even through the two concepts are not equal, we can still observe that the graph transformer outperforms the GCN in almost all scenarios.
>
> * **Q2**: The illustration in Figure 3 and the corresponding motivation are very similar to [1]. The paper lacks a comparison with [1].
> * * **A2**: Thanks for the kindly remind for the most recent work [1] published at ICLR 2025. Both our work and [1] have referenced the non-isomorphic MILP example shown in figure 2 in [2]. However, the motivation of our work and [1] are total different. The motivation of [1] attempts to solve the expressive issue of graph neural networks for representing MILP proposed in [2] under GNN framework. Our design jumps out scope of GNN framework. As far as we known, we are the first to introduce the transformer into the half-convolutions to learn the representation of the MILP. It solves both the expressive issue and the limited receptive field issue for GNN efficiently. We provide both theoretical analysis and experimental results to demonstrate the clear advantages of our method over previous frameworks. Furthermore, our work does not limited to the representation issues. We also considered the combination of reinforcement learning and imitation learning that provide better training for graph-transformers.
> * **Q3**: Why is the primal gap of CL-LNS so different from the original paper? For example, the original CL-LNS paper shows a primal gap of less than 1%, but it is 33% and 72% in this paper on the same problems — CA and SC.
> * * **A3**: CL-LNS have used the ECOLE library, but due to the ECOLE library no longer being maintained (last updated in 2022), we are unable to successfully install many dependent libraries and thus cannot run it directly on our datasets. Therefore, we have to reproduce it based on the framework and algorithmic details described in the paper. In addition, to provide a more fair comparison, we have to unify the features and problems for all baselines, which might also have negative impact on CL-LNS.
>
> [1] Ye, Xinyu, et al. "On Designing General and Expressive Quantum Graph Neural Networks with Applications to MILP Instance Representation." The Thirteenth International Conference on Learning Representations. 2025.
>
> [2] Ziang Chen, Jialin Liu, Xinshang Wang, and Wotao Yin. On representing mixed-integer linear programs by graph neural networks. In The Eleventh International Conference on Learning Representations, 2023.

---

> ### Author Response · Authors · 2025-06-06
> **Response to Reviewer 66od (part 2 of 2)**
>
> * **Q4**: The metrics used in this paper are uncommon in this field, where primal gap and primal integral are typically reported. What does objective improvement mean? Why are they not zero for optimal solutions?
> * * **A4**: Thank you for your suggestion. In the previous version of the paper, we used the term *GAP* in the table , which is actually the *primary gap*. To avoid confusion, we will rename it to primary gap (PG) in the latest version of the paper. In fact, we only consider the PG in our paper as we follow some previous work like [3]. As the importance of primal intergral (PI), we have added additional experiments to evaluate it in the latest version. The additional experiments show that our method also have the best PI, which is consistent with the results observed using PG. We will report it in the latest version of the paper.
> * * The *objective improvement* is defined as the difference between the current objective function value and the initial objective function value to show the improvement of the MILP. Therefore, it is not zero for optimal solutions. Please refer to *5.2.2 Metrics* in the latest version of the paper for a detailed definition.
>
> * **Q5**: The paper lacks a comparison with Wu et al., 2021, which also uses RL to train the LNS.
> * * **A5**: Thanks for the kindly suggestion. We have been working diligently to deploy the code from Wu et al. However, its implementation relies on the ECOLE library, which is no longer maintained (last updated in 2022). As a result, we encountered numerous issues installing its dependencies. The baseline CL-LNS also depends on the ECOLE library and face the same challenges. We take much time to rewrite the code according to the framework and algorithmic details described in the paper. Due to the time limitation, we cannot rewrite the code in Wu et al and realize it as the original one.
>
> * **Q6**: What is the inference time of the proposed method compared to other baselines? How many the improvement steps are applied during inference?
> * * **A6**: We add the running time experiment and compare the per-step inference time of our method with other baselines shown in table 7 in the latest version. It shows that our method is only slightly slower than the fastest baseline and much faster than the rest baselines. Although the graph transformer is relatively more complex than the models used in other works, it does not dominate the inference time. The inference also include the sub-problems solving time, which is set to be 5 seconds for SC and CA, 10 seconds for IP. We set the improvement steps during inference at most 15 steps. However, we find that LNS converge quickly at around 4 steps, which is consistent with the training step setting. It indicates that our method can also have fast total inference time.
>
> * **Q7**: What is the impact of the weight introduced in Equation (5)? How to balance the RL loss and the imitation loss?
> * * **A7**: The weight in Equation (5) measures the quality of each sample. The samples with better objective, i.e., smaller gap to the optimal objective, should be assigned higher weights. It assists the expert to learn from more important samples. We balance the RL loss and imitation loss in Equation(16) by a hyper-parameter $\alpha$.
>
> * **Q8**: The imitation targets obtained from local branching introduce additional computational overhead.
> * * **A8**: The imitation targets (i.e., expert policy) obtained from local branching indeed introduce extra computational overhead. However, this computational is one-time calculation before the training of RL. When compared with the training time of RL, learning the imitation targets can be ignored. In addition, with the guidance of imitation targets, the RL can converge faster during the training, which might reduce the total computation time.
>
> * **Q9**: The generalization performance of the method is unclear. Would be better to test it on benchmark datasets like Distributional MIPLIB, which includes MILP instances with different distributions and levels of hardness.
> * * **A9**: In the latest version of the paper, we have added a generalization analysis shown in Table 4-6 in the new subsection 5.4.3. The results show that the graph transformer indeed enhances the generalization ability of RL. It assists our RL-based LNS method in having good generalization, which improves its adaptability for samples with different sizes during testing.
>
> [3] Yaoxin Wu, Wen Song, Zhiguang Cao, and Jie Zhang. Learning large neighborhood search policy for integer programming. In A. Beygelzimer, Y. Dauphin, P. Liang, and J. Wortman Vaughan (eds.), Advances in Neural Information Processing Systems, 2021.

---

> > ### Comment · Reviewer_66od · 2025-06-08
> > **Still have some concerns**
> >
> > Thanks for the response. After carefully reading the rebuttal and other reviewers’ comments, I still have the following concerns, which I believe are non-negligible:
> >
> > 1) The implementation of the baselines is questionable. Although the author(s) claimed that they have reproduced the baselines (IL-LNS, CL-LNS) based on the framework and algorithmic details described in the original paper, the results differ significantly from those in the original paper, making it impossible to compare the proposed method with the baselines. This concern is also mentioned by Reviewer dgVS.
> >
> > 2) Wu et al. (2021), which uses RL to train LNS, is highly relevant and should be included, as this paper uses IL+RL to learn LNS. This is also mentioned by Reviewer f8YJ.
> >
> > 3) The method of calculating the weight in Equation (5) is ambiguous. Is the weight based on the ratio between the current and old policy probabilities, or is it based on the reward? The notation "r" is used ambiguously. Additionally, why should it be calculated in the softmax way? Do you have any ablations on it?
> >
> > 4) The selection of the hyperparameter \alpha seems vague to me.  As the author(s) mentioned in the response to Reviewer f8YJ (Part 2), "even if starting with a reasonably good policy, the agent's performance deteriorated after a period of training". I believe the balance between actor exploration (RL) and expert guidance (IL) is critical for the training. If the IL loss weight is too large, the policy may overly mimic the expert; if too small, it may drift due to unstable RL training. It would be more convincing if the author(s) could provide an in-depth discussion on it with additional empirical results.

---

> > > ### Author Response · Authors · 2025-06-12
> > > **Response to Reviewer 66od (part 1 of 2)**
> > >
> > > Many thanks for your feedback and enthusiasm about our paper.
> > >
> > > We would like to make the following comments:
> > >
> > > * **Q1.** The implementation of the baselines is questionable. Although the author(s) claimed that they have reproduced the baselines (IL-LNS, CL-LNS) based on the framework and algorithmic details described in the original paper, the results differ significantly from those in the original paper, making it impossible to compare the proposed method with the baselines. This concern is also mentioned by Reviewer dgVS.
> > >
> > > * * **A1.** First, the scale of the instances we use differs from those in the original paper. Since PG is a relative measure, its value varies depending on the instance scale and the reference algorithm used for comparison. In the original CL-LNS paper, the reported PG was below 1%, whereas [1] computes PB using their proposed algorithm, yielding values between 1% and 10%. In our work, we calculate PG based on the optimal solution, which may result in numerical differences from the original paper. In addition, the unsatisfactory performance of CL-LNS in our experimental results has also been observed in other works. For example, the results in [2] show that IL-LNS generally outperforms CL-LNS in most cases. Similarly, [3] reports comparable performance between R-LNS and CL-LNS, which aligns with our findings.
> > >
> > > * **Q2.** Wu et al. (2021), which uses RL to train LNS, is highly relevant and should be included, as this paper uses IL+RL to learn LNS. This is also mentioned by Reviewer f8YJ.
> > >
> > > * * **A2.** In fact, we found that the agent could barely learn using reinforcement learning alone. The experimental results in [4] also shows that RL-LNS generally performs worse than IL-LNS and CL-LNS, and even underperforms compared to Random in CA-S and MVC-L. Similarly, in [1], RL-LNS is outperformed by CL-LNS in all cases except MC-2000.
> > >
> > > * **Q3.** The method of calculating the weight in Equation (5) is ambiguous. Is the weight based on the ratio between the current and old policy probabilities, or is it based on the reward? The notation "r" is used ambiguously. Additionally, why should it be calculated in the softmax way? Do you have any ablations on it?
> > >
> > > * * **A3.** The idea behind Equation (5) is similar to the approach used in [5] (Equations (11) and (12)), which construct the conditional distribution with objective values via energy functions. A similar technique is also applied in [6] (Equation (4)). However, unlike these methods, which focus on predicting complete solutions for instances with binary variable values (0 or 1), our approach is designed for variable selection rather than full solution prediction.
> > >
> > >
> > >
> > > [1] Kong, S., Liu, C. &amp; Gomes, C.. (2024). ILP-FORMER: Solving Integer Linear Programming with Sequence to Multi-Label Learning. Proceedings of the Fortieth Conference on Uncertainty in Artificial Intelligence, in Proceedings of Machine Learning Research 244:2018-2028 Available from https://proceedings.mlr.press/v244/kong24a.html.
> > >
> > > [2] Shengyu Feng, Shengyu_Feng, Zhiqing Sun, Yiming Yang. (2025). Sampling-Enhanced Large Neighborhood Search for Solving Integer Linear Programs.
> > >
> > > [3] Wenbo Liu, Akang Wang, Wenguo Yang and Qingjiang Shi. (2024). Mixed-Integer Linear Optimization via Learning-Based Two-Layer Large Neighborhood Search. arXiv preprint arXiv:2412.08206.
> > >
> > > [4] Taoan Huang, Aaron Ferber, Yuandong Tian, Bistra Dilkina, and Benoit Steiner. (2023) Searching large neighborhoods for integer linear programs with contrastive learning.
> > >
> > > [5] Vinod Nair, Sergey Bartunov, Felix Gimeno, Ingrid von Glehn, Pawel Lichocki, Ivan Lobov, Brendan O’Donoghue, Nicolas Sonnerat, Christian Tjandraatmadja, Pengming Wang, Ravichandra Addanki, Tharindi Hapuarachchi, Thomas Keck, James Keeling, Pushmeet Kohli, Ira Ktena, Yujia Li, Oriol Vinyals, and Yori Zwols. (2020). Solving mixed integer programs using neural networks.
> > >
> > > [6] Qingyu Han and Linxin Yang and Qian Chen and Xiang Zhou and Dong Zhang and Akang Wang and Ruoyu Sun and Xiaodong Luo. (2023). A GNN-Guided Predict-and-Search Framework for Mixed-Integer Linear Programming. arXiv preprint arXiv:2302.05636.

---

> > > ### Author Response · Authors · 2025-06-12
> > > **Response to Reviewer 66od (part 2 of 2)**
> > >
> > > * **Q4.** The selection of the hyperparameter \alpha seems vague to me. As the author(s) mentioned in the response to Reviewer f8YJ (Part 2), "even if starting with a reasonably good policy, the agent's performance deteriorated after a period of training". I believe the balance between actor exploration (RL) and expert guidance (IL) is critical for the training. If the IL loss weight is too large, the policy may overly mimic the expert; if too small, it may drift due to unstable RL training. It would be more convincing if the author(s) could provide an in-depth discussion on it with additional empirical results.
> > > * * **A4.** The balance between actor exploration and expert guidance is critical. In fact, expert guidance plays a particularly important role in our algorithm. Without expert guidance—or even when it is provided in equal proportion to reinforcement learning—the agent struggles to learn effectively. A higher proportion of expert guidance helps prevent the training strategy from diverging too far from the expert's behavior. Therefore, IL loss weight is larger than that of reinforcement learning. Due to the high computational cost of reinforcement learning, we plan to include a detailed experimental analysis on this aspect in the final version of the paper.

---

### Review · Reviewer_dgVS · 2025-05-22

**Summary Of Contributions:**

This paper introduces a new method to solve MILPs by learning destroy operators for the Large Neighborhood Search (LNS). The contribution is two-fold: a weighted expert-guided training method based on RL, and a graph transformer with linear attention to process the MILP instances.

**Audience:**

Yes

**Claims And Evidence:**

Yes

**Requested Changes:**

### Critical

1. Specify the value of $\alpha$ for the expert guidance.

2. Justify why we should use linear attention. What if we replace it with normal attention? Can it generate better solutions, and if so in how much time?

3. Please report some primal integral values against baselines as it is hard to judge the anytime performance.


### Minor

1. There are several typos throughout the paper e.g. “we take usage of redundant”, “tranfsormer”, “Gruobi”, “In this session” “coefffcient” to name a few.

2. The code provided does not open and results in an error.

**Strengths And Weaknesses:**

### Strengths

The paper studies an important topic, relevant for many industrial applications. Overall the manuscript is well-written, and the contribution regarding training method is novel. The experiments are performed on relevant problems and demonstrate the efficacy of the approach.

### Weaknesses

1. The graph transformer architecture novelty is limited: in fact, CL-LNS uses a GAT already instead of a GCN, which I guess also satisfies the proposed theorem. What are the differences with the one proposed in CL-LNS?

2. The use of linear attention was not justified fully: why not use full attention instead? Problems in the range of variables considered are well within the capabilities of modern hardware, and quadratic attention has much more expressivity.

3. The positional encoding via shortest path is too expensive; the authors mention 40 minutes to compute it (per problem or for all of them?). This would not justify the proposed network in practical settings. Also, the PE seems to be the main “secret sauce” according to Table 3, but also the most expensive component by far.

4. The choice of experiments diverges noticeably from prior works such as IL-LNS and CL-LNS in both problems and reported numbers (i.e. gaps in double digits). Why was a different setting chosen?

5. Similar to point 4), Primal Integral (PI) was not reported. For LNS, this should be a more important metric.

---

> ### Author Response · Authors · 2025-06-06
> **Response to Reviewer dgVS (part 1 of 2)**
>
> Many thanks for your feedback and enthusiasm about our paper. We now comment on the requested changes and weaknesses.
>
> We now address the weakness you listed:
> * **Q1**. The graph transformer architecture novelty is limited: in fact, CL-LNS uses a GAT already instead of a GCN, which I guess also satisfies the proposed theorem. What are the differences with the one proposed in CL-LNS?
> * * **A1**. Both GCN and GCN belong to the Message Passing Neural Network(MPNN) framework. It has been proven that all MPNN framework is, at most, as expressive as the 1-WL Test. Therefore, GAT does not satisfy the proposed theorem, i..e, it is at most expressive with 1-WL test. Our Graph transformer overcome the expressive issue faced by MPNN framework. In addition, both GCN and GCN suffer from a limited receptive field, while capturing global information from graphs is very important for MILP. Our graph transformer is good at capturing the global information. In addition, as far as we known, we are the first work introducing the transformer into the half-convolutions to learn the representation of the MILP. It jump out scope of MPNN framework and solves both the expressive issue and the limited receptive field issue efficiently. We provide both theoretical analysis and experimental results to demonstrate the clear advantages of our method over previous frameworks.
>
> * **Q2**. The use of linear attention was not justified fully: why not use full attention instead? Problems in the range of variables considered are well within the capabilities of modern hardware, and quadratic attention has much more expressivity.
> * * **A2**. Linear attention and quadratic attention have been discussed in [1]. Firstly, the computation complexity for linear attention can be achieved in $\mathcal{O}(N)$, which is much more efficient than the Softmax attention in original Transformers that requires $\mathcal{O}(N^{2})$. While the Softmax attention possesses provable expressivity, its quadratic complexity hinders the scalability for graphs. Linear attention reduces the quadratic complexity to $\mathcal{O}(N)$ and in the meanwhile guarantee the expressivity for learning all-pair interactions. Besides, the experimental results in [1] show that the linear attention performs better than GCN, GAT, and quadratic attention on datasets such as Cora and CiteSer. Considering that reinforcement learning itself consumes a significant amount of time and space, we believe that using linear attention is more appropriate.
> * **Q3**. The positional encoding via shortest path is too expensive; the authors mention 40 minutes to compute it (per problem or for all of them?). This would not justify the proposed network in practical settings. Also, the PE seems to be the main “secret sauce” according to Table 3, but also the most expensive component by far.
> * * **A3**. Our position encoding is pre-calculated and only needs to be computed once before training begins. The position encoding is not the most expensive component. The training of expert policy and the training of RL all take much more time than position encoding. In fact, the training of RL is the most expensive component, which usually requires more than ten hours.
> In addition, the process of position encoding can be optimized and the time can be reduced significantly. The current method in our paper spends 40 minutes because we choose the networkx library and Dijkstra's algorithm to calculate the shortest path for all pairs. For a graph with non-negative edge weights, the time complexity of solving the single source shortest path problem using Dijkstra's algorithm is $\mathcal{O}(n^2)$, combined with the Fibonacci heap, the time complexity can reach $\mathcal{O}(m+n \ log n)$, and the time complexity of solving the all-pairs shortest path problem is $O (mn+n^2\log n)$. In fact, for sparse graphs, there are currently faster algorithms to calculate the shortest path for all pairs, such as the Contraction Hierarchies  algorithm. The shortest path position encoding for 300 problems containing 1500 nodes and 1600 constraints can be calculated in 5 minutes. We will provide additional explanations in the latest version of the paper.
>
> [1] Qitian Wu, Wentao Zhao, Chenxiao Yang, Hengrui Zhang, Fan Nie, Haitian Jiang, Yatao Bian, and JunchiYan. Sgformer: Simplifying and empowering transformers for large-graph representations. In Advances in Neural Information Processing Systems (NeurIPS), 2023.

---

> ### Author Response · Authors · 2025-06-06
> **Response to Reviewer dgVS (part 2 of 2)**
>
> * **Q4**. The choice of experiments diverges noticeably from prior works such as IL-LNS and CL-LNS in both problems and reported numbers (i.e. gaps in double digits). Why was a different setting chosen?
> * * **A4**. CL-LNS have used the ECOLE library, but due to the ECOLE library no longer being maintained (last updated in 2022), we are unable to successfully install many dependent libraries and thus cannot run it directly on our datasets. Therefore, we have to reproduce it based on the framework and algorithmic details described in the paper. In addition, to provide a more fair comparison, we have to unify the features and problems for all baselines, which might also have negative impact on CL-LNS.
>
> * **Q5**. Similar to point 4), Primal Integral (PI) was not reported. For LNS, this should be a more important metric.
> * * **A5**. Thank you for your suggestion. In fact, we only consider the Primal Gap (PG) in our paper as we follow some previous work like [2]. As the importance of primal intergral (PI), we have added additional experiments  to evaluate it in the latest version of the paper. The additional experiments show that our method also have the best PI, which is consistent with the results observed using PG. We will report it in the latest version of the paper.
>
> We now address the requested changes you listed:
> * **C1**. Specify the value of $\alpha$ for the expert guidance.
> * * **R1**. Thank you for your suggestion. In the latest version of the paper, we will add the value of $\alpha$ in *5.2.2 Hyper-parameters*.
> * **C2**. Justify why we should use linear attention. What if we replace it with normal attention? Can it generate better solutions, and if so in how much time?
> * * **R2**. Please refer to the reply to Q2 and we would add the discussion on the latest version.
> * **C3**. Please report some primal integral values against baselines as it is hard to judge the anytime performance.
> * * **R3**. Please refer to the reply to Q5 and we have add the PI into our experiment in the latest version.
> * **C4**. There are several typos throughout the paper e.g. “we take usage of redundant”, “tranfsormer”, “Gruobi”, “In this session” “coefffcient” to name a few.
> * * **R4**. Thanks for your patient. We have revised these typos in the latest version.
> * **C5**. The code provided does not open and results in an error.
> * * **R5**. We have rechecked our code link and it is feasible.
>
>
> [2] Yaoxin Wu, Wen Song, Zhiguang Cao, and Jie Zhang. Learning large neighborhood search policy for integer programming. In A. Beygelzimer, Y. Dauphin, P. Liang, and J. Wortman Vaughan (eds.), Advances in Neural Information Processing Systems, 2021.

---

### Review · Reviewer_f8YJ · 2025-05-23

**Summary Of Contributions:**

The paper presents a new approach for neural large neighbourhood search (LNS) in mixed integer linear program via Graph Transformers trained through a combination of RL and expert guidance. Specifically the approach includes: (1) a graph transformer architecture that includes two global attention units and a GCN layer that is shown to be more expressive than 1-WL test; (2) an actor-critic training procedure that incorporate weighted expert guidance during training. Experiments on three types of problems -- set covering, combinatorial auction, and balanced item placement -- show significant gains over the baselines.

**Audience:**

Yes

**Broader Impact Concerns:**

I have no ethical concerns regarding the paper.

**Claims And Evidence:**

No

**Requested Changes:**

**Extending the experiments:**
  - Include Wu et al. as baseline
  - More detailed ablation study as mentioned above
  - Analysis on generalization to larger instances
  - Clarification on time limit and analysis for both shorter and longer times consistent with previous literature
  - Clarify the timing of the position encoding computation

**Methodology:**
- Clarify notation as mentioned above
- [Not critical] compare to simpler incorporation of expert guidance via pre-training and RL fine-tuning

**Existing literature:**
- Compare with [1] -- perhaps not critical

**Strengths And Weaknesses:**

## Strengths:
- Using neural network for solving combinatorial optimization problems is an important research area that has received significant attention in recent years
- New approach for LNS based on Graph Transformer trained using a combination of RL and expert guidance
- Experiments show significant gain over baselines
- Paper is clear and well-written

----
## Weaknesses:

**Experiments:** my main concerns are regarding the experiments:
- Wu et al. (mentioned in the paper as the only RL-based approach) is not used as baseline and should be considered.
- Ablations could be more extensive to account for the various design choices. For example instead of w/o RL we can also consider w/ RL but w/o expert guidance
- Generalization to larger instance: consistent with literature on NN for combinatorial optimization broadly, including relevant previous work for LNS like Wu et al., experiments should consider both testing on instances of the same size as the training instances as well as testing on larger instances to evaluate generalization to larger problems. At the moment, based on what I understood from the paper, it seems that test instances are only ~30% larger than training instances. This is quite limited scaling. In contrast, Wu et al., and recently Yuan et al. [1], have tested on problems up to four-time larger than training problems in terms of the number of variables and constraints.
- Time limit: it is not clear how the comparison between the methods account for runtime. While all approaches seem to be limited to step limit of 4, the time required for one step can significantly vary between approaches leading to unfair comparison. Previous work (e.g., Wu et al.) has set up a time limit and compared the performance at the time limit to provide fair comparison. Finally, the experiments should not be conducted for a single time/step limit, but looking at least on two limits (shorter and longer).
- The paper notes that it requires 40 minutes to compute position encoding. Is this the amount of time for 300 instances (i.e., the average time per instance is 40/300)?

**Methodology:**
- Notation: what is $N$ in Eq. 7 and 8 and in O(N)?
- I could not find a discussion or motivation for the chosen approach for inclusion of expert guidance vs. a simpler approach of pre-training the policy using cross entropy and then fine-tuning using RL? Does it perform better this way?

**Comparison with literature:** a recent highly-relevant work is not included [1], however this is a recent work and perhaps considered contemporary so I ignored it in my evaluation and leave this to the area editor.

**Typos/Grammer:**
- "design an variable-pair” -> “design a variable-pair” (in 4.3.1)
- "are capture via the” ->  "are captured via the”


[1] Yuan, Hao, et al. "BTBS-LNS: Binarized-Tightening, Branch and Search on Learning LNS Policies for MIP." The Thirteenth International Conference on Learning Representations.

---

> ### Author Response · Authors · 2025-06-06
> **Response to Reviewer f8YJ (part 1 of 2)**
>
> Thank you for your valuable questions. We would like to make the following comments:
>
> About extending the experiments:
> * **Q1**. Include Wu et al. as baseline.
> * * **A1**. Thanks for the kindly suggestion. We have been working diligently to deploy the code from Wu et al. However, its implementation relies on the ECOLE library, which is no longer maintained (last updated in 2022). As a result, we encountered numerous issues installing its dependencies. The baseline CL-LNS also depends on the ECOLE library and face the same challenges. We take much time to rewrite the code according to the framework and algorithmic details described in the paper. Due to the time limitation, we cannot rewrite the code in Wu et al and realize it as the original one.
>
> * **Q2**. More detailed ablation study as mentioned above.
> * * **A2**. Thanks for the kindly suggestion. Although not mentioned in the paper, we actually consider the following two methods: (1) w/RL but w/o expert guidance and (2) w/RL with expert network as initialization. The results show that in both cases, the agent could barely learn. Even if starting with a reasonably good policy, the agent's performance deteriorated after a period of training. Solving the subproblems failed to improve the original problem (i.e., reward=0), leading to the failure of RL training. These results indicate that training purely with reinforcement learning is extremely challenging in this context. The experts in our RL-based method indeed assist in stabilizing the training process.
>
> * **Q3**. Analysis on generalization to larger instances.
> * * **A3**. Thanks for the good suggestion. In the latest version of the paper, we have added a generalization analysis. We test the same/twice/triple size as the training instances. The results show that the graph transformer indeed enhances the generalization ability of RL. It assists our RL-based LNS method in having good generalization, which improves its adaptability for samples with different sizes during testing. Please refer to the additional subsection *5.4.3 Generalization analysis* in the latest paper for more details.
>
> * **Q4**. Clarification on time limit and analysis for both shorter and longer times consistent with previous literature.
> * * **A4**. Thank you for your suggestion. In order to fairly compare all methods, we select a step size limit of 4 during the inference, which is the same with the training step. The time limit for Gurobi to solve the sub-problems in each step is 5 seconds for SC and CA, and 10 seconds for IP. We add the running time experiment and compare the per-step inference time of our method with other baselines shown in table 7 in the latest version. It shows that our method is only slightly slower than the fastest baseline and much faster than the rest baselines. Please refer to the additional subsection *5.4.4 Efficiency analysis* in the latest paper for more details.
>
> * **Q5**. Clarify the timing of the position encoding computation.
> * * **A5**. Our position encoding is pre-calculated aand only needs to be computed once before training begins. The training of expert policy and the training of RL all take much more time than position encoding. In fact, the training of RL is the most expensive component, which usually requires more than ten hours. When comparing with it, the time of calculating position encoding can be ignored.
> In addition, the process of position encoding can be optimized and the time can be reduced significantly. The current method in our paper spends 40 minutes because we choose the networkx library and Dijkstra's algorithm to calculate the shortest path for all pairs. For a graph with non-negative edge weights, the time complexity of solving the single source shortest path problem using Dijkstra's algorithm is $\mathcal{O}(n^2)$, combined with the Fibonacci heap, the time complexity can reach $\mathcal{O}(m+n \ log n)$, and the time complexity of solving the all-pairs shortest path problem is $O (mn+n^2\log n)$. In fact, for sparse graphs, there are currently faster algorithms to calculate the shortest path for all pairs, such as the Contraction Hierarchies  algorithm. The shortest path position encoding for 300 problems containing 1500 nodes and 1600 constraints can be calculated in 5 minutes. We will provide additional explanations in the latest version of the paper.

---

> ### Author Response · Authors · 2025-06-06
> **Response to Reviewer f8YJ (part 2 of 2)**
>
> About methodology:
> * Clarify notation as mentioned above.
> * * $N$ is the number of nodes.  For 	variable-pair (constraint-pair) attention, its the number of variables (constraints). We will clarify this in the paper.
> * Compare to simpler incorporation of expert guidance via pre-training and RL fine-tuning.
> * * We actually consider the method with pre-training and then RL fine-tuning. We find that the agent could barely learn. Even if starting with a reasonably good policy, the agent's performance deteriorated after a period of training. Solving the subproblems failed to improve the original problem (i.e., reward=0), leading to the failure of RL training. It is very difficult to train only using reinforcement learning even with good initialization. The expert guidance in our RL-based method indeed assist in stabilizing the training process.
>
>
> About Existing literature：
> * Compare with [1].
> * * Thanks for the kindly remind for the most recent work [1] published in ICLR 2025. The problem addressed by the references is not aligned with our focus. [1] mainly focuses on solving general integer programming problems, while we pay more attention to new network architectures and training methods. In addition, for it is relatively new and the code is not yet open source, it is difficult to reproduce and compare the results.
>
> About typos/grammer:
> * * Thanks for your patient. We revised these typos in the latest version.
>
> [1] Yuan, Hao, et al. "BTBS-LNS: Binarized-Tightening, Branch and Search on Learning LNS Policies for MIP." The Thirteenth International Conference on Learning Representations.

---

### Decision · Action_Editor_eCdb · 2025-07-20

**Recommendation:** Reject

**Audience:**

Yes

**Audience Explanation:**

The proposed method in this manuscript is interesting, and there could certainly be an audience for it. The authors will need a major revision to address the soundness of the experiment baselines, so the audience will get a clear picture of the proposed method's performance.

**Claims And Evidence:**

No

**Claims Explanation:**

The choice of experiments diverges noticeably from prior works such as IL-LNS and CL-LNS in both problems and reported numbers (i.e. gaps in double digits). The gap is way too large and is beyond a reasonable discrepancy in replicating an existing work. This large gap challenges the soundness of the claims made by the submission.

**Resubmission Of Major Revision:**

The authors may consider submitting a major revision at a later time.